# The M$_w$ 7.5 Tadine (Maré, Loyalty Is.) earthquake and related tsunami of December 5, 2018: seismotectonic context and numerical modelling

Jean Roger[1,2*], Bernard Pelletier[3], Maxime Duphil[1], Jérôme Lefèvre[1], Jérôme Aucan[1], Pierre Lebellegard[3], Bruce Thomas[1,4], Céline Bachelier[5], David Varillon[5]

[1]ENTROPIE, Institut de Recherche pour le Développement, 101, Promenade Roger Laroque, BP A5 98848 Nouméa CEDEX, New Caledonia

[2]Now at: GNS Sciences, 1 Fairway Drive, Lower Hutt 5010, New Zealand

[3]GEOAZUR, Institut de Recherche pour le Développement, 101, Promenade Roger Laroque, BP A5 98848 Nouméa CEDEX, New Caledonia

[4]LISAH, Univ Montpellier, INRAE, IRD, Institut Agro, Montpellier, France

[5]IMAGO, Institut de Recherche pour le Développement, 101, Promenade Roger Laroque, BP A5 98848 Nouméa CEDEX, New Caledonia

*Correspondence to:* J. Roger (j.roger@gns.cri.nz)

**Abstract.** On the 5$^{th}$ of December 2018, a magnitude M$_w$ 7.5 earthquake occurred southeast of Maré, an island of the Loyalty Archipelago, New Caledonia. This earthquake is located at the junction between the plunging Loyalty Ridge and the southern part of the Vanuatu Arc, in a tectonically complex and very active area regularly subjected to strong seismic crises and earthquakes higher than magnitude 7 and up to 8. Widely felt in New Caledonia, it has been immediately followed by a tsunami warning, confirmed shortly after by a first wave arrival at the Loyalty Islands tide gauges (Maré and Lifou), then along the east coast of Grande Terre of New Caledonia and in several islands of the Vanuatu Archipelago. Two solutions of the seafloor initial deformation are considered for tsunami generation modelling, one using a non-uniform finite source model from USGS, and the other being a uniform slip model built from the GCMT solution, the geological knowledge of the region and empirical laws establishing relationships between the moment magnitude and the fault plane geometry. Both tsunami generation and propagation are simulated using SCHISM, an open-source modelling code solving the shallow water equations on an unstructured grid allowing refinement in many critical areas. The results of numerical simulations are compared to tide gauge records, field observations and testimonials from 2018. Careful inspection of wave amplitude and wave energy maps for the two simulated scenarios shows clearly that the heterogeneous deformation model is inappropriate, while it raises the importance of fault plane geometry and azimuth on tsunami amplitude and directivity. While the arrival times, wave amplitude and polarities obtained with the uniform slip model are globally coherent, especially in far-field locations (Hienghène, Poindimié and Port Vila). Due to interactions between the tsunami waves and the numerous bathymetric structures like the Loyalty and Norfolk Ridges in the neighborhood of the source, the tsunami propagating toward the south of Grande Terre and the Isle of Pines is captured by these structures acting like waveguides, allowing it to propagate to the north-northwest, especially in the Loyalty Islands and along the east coast of Grande Terre. A similar observation results from the propagation in the Vanuatu islands, from Aneityum to Efate.

## 1 Introduction

At 04:18:08 UTC on December 5, 2018 (15:18:08 local time in New Caledonia – UTC+11), a major earthquake of magnitude $M_w7.5$ occurred 165 km east-south-east of Tadine, Maré, the southernmost inhabited island of the Loyalty Archipelago (Figure 1). It was strongly felt in New Caledonia (Loyalty Islands and the Grande Terre) as far as Nouméa, more than 300 km west from the source (Roger et al., 2019a, 2019b, 2019c), while the effects were weaker in Port Vila, capital of Vanuatu, about 470 km to the North according to a CBS News interview of Mr. McGarry, media director at the Vanuatu Daily Post. There is no report of damage linked to the earthquake.

Within minutes, its location and magnitude were determined by the Seismological Observatory of New Caledonia (http://www.seisme.nc, https://bit.ly/2IMkmgM) [$M_w7.6$, 22.01°S, 169.33°E, 30 km], by USGS [$M_w7.5$, 21.968°S, 169.446°E, 10 km] and by the Global CMT project as a quick CMTS [$M_w7.5$, 21.95°S, 169.25°E] (Dziewonski et al., 1981; Ekström et al., 2012). Maximum distance between these three locations is ~15 km, in agreement with the acceptable location errors between the different observatories. The revised hypocenter location of the event provided by USGS, GCMT, and GEOSCOPE is respectively 21.950°S, 169.427°E, 10 km, 21.95°S, 169.25°E, 17.8 km and 21.969°S, 169.446°E, 12km.

The seismic moment $M_o$ of this event has been evaluated to 2.49 x $10^{20}$ N.m ($M_w7.53$) by USGS, 2.52 x $10^{20}$ N.m by GCMT project ($M_w7.5$), and 2.95 x $10^{20}$ N.m ($M_w7.58$) by the SCARDEC method (GEOSCOPE-IPGP).

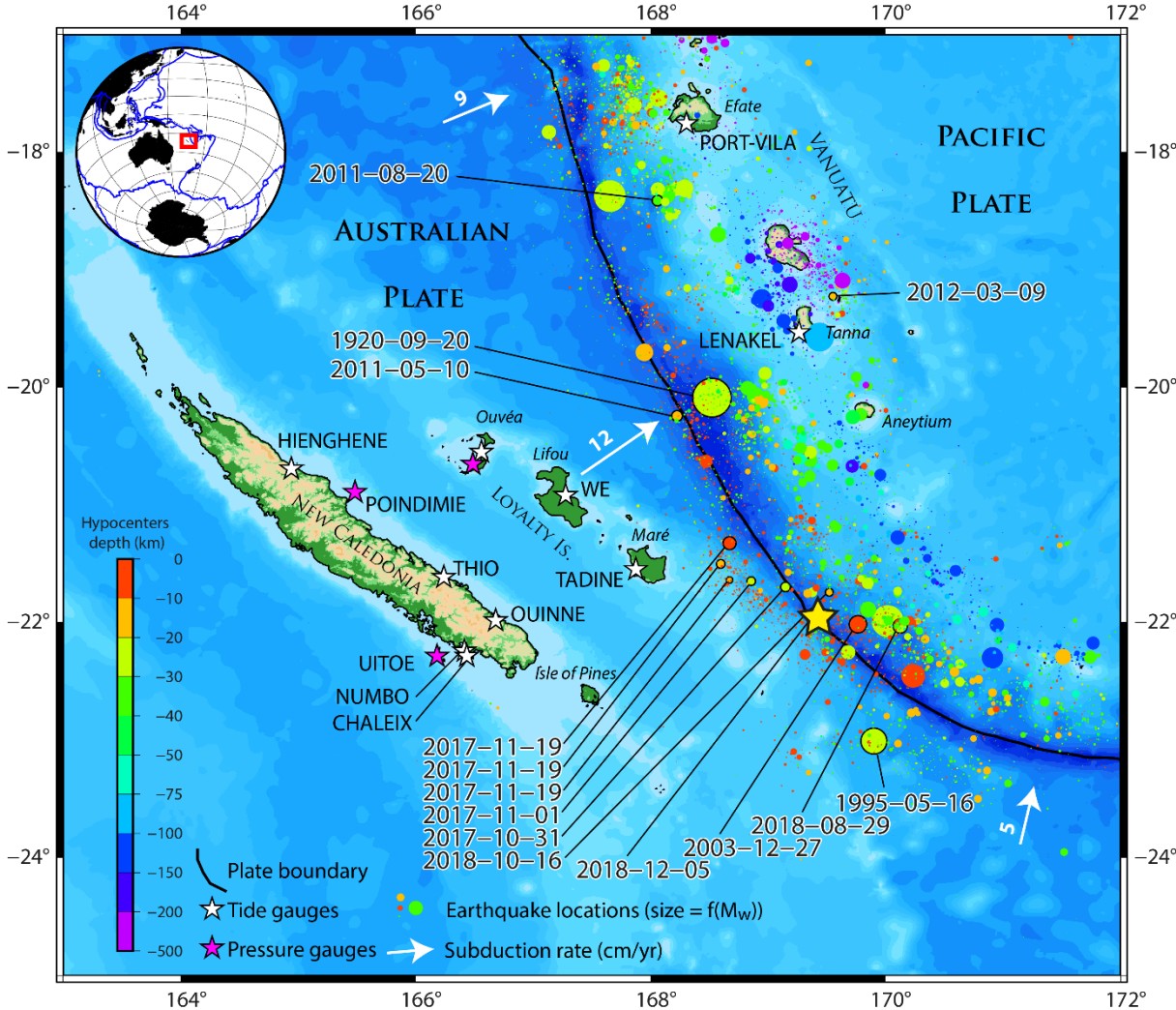

**Figure 1 : New Caledonia and the south Vanuatu Subduction Zone. The colored dots represent the seismicity from the USGS database for the period from January 1, 1900 to January 24, 2019, with the size of dots being proportional to event's magnitude. Tsunamigenic earthquakes having been recorded in New Caledonia (Roger et al., 2019b) are highlighted with a black circle and linked to dates. The white arrows symbolize the subduction directions and rates of the subducting Australian Plate under the Pacific plate. Tide and pressure gauges able to record tsunami waves are respectively symbolized with white and purple stars. The yellow star locates the December 5, 2018 earthquake's epicenter. The study area is located within the red rectangle in the southwestern Pacific Ocean.**

Considering the strong magnitude of this shallow earthquake, a tsunami alert was released locally by the IRD seismological laboratory to the New Caledonia civil security service (DSCGR) and regionally by the NOAA/PTWC shortly after the earthquake occurred. A tsunami was confirmed by real-time tide gauges measurements within minutes at first in the Loyalty Islands, then in most places of the New Caledonia/Vanuatu region.

This earthquake is added to the list of local earthquakes reported by the past in the south Vanuatu Subduction Zone and especially to the two shocks that triggered major tsunamis in the Loyalty Islands in March 28, 1875 and September 20, 1920 (Sahal et al., 2010) with estimated magnitude of 8.1-8.2 and 7.5-7.8 respectively (Ioualalen et al., 2017), and with the $M_w$7.7 May 17, 1995 event which occurred close, and south, to the

December 5, 2018 event showing a similar focal mechanism (i.e. normal faulting in the plunging plate), as explained hereunder. The tsunami generated by the 1995 earthquake was well observed at the entrance of the first lagoon and on Erakor Island in Port Vila, located south of Efate, Vanuatu (Lardy, 1995).

This study aims to (1) simulate the December 5, 2018 tsunami in New Caledonia and Vanuatu, comparing the computed maximum amplitudes and the synthetic waveforms to those observed and/or recorded on tide gauges and (2) discuss the role of earthquake source parameters through sensitivity tests. The first part of the article deals with the particular seismotectonic context of the region between New Caledonia and Vanuatu and its ability to trigger tsunamigenic earthquakes. The second and third parts focus respectively on the December 5, 2018 tsunami observations and records and tsunami numerical modelling and the fifth and sixth ones present and discuss the simulation results and provide study prospects.

## 2 Seismotectonic context

### 2.1 Junction of the Loyalty Ridge and the Vanuatu subduction zone

The December 5, 2018, $M_w$ 7.5 earthquake is located southeast of Maré (Loyalty Islands, New Caledonia), immediately west of the southern part of the Vanuatu (former New Hebrides) trench in the junction area between the Loyalty Ridge and the Vanuatu Arc (Figure 1). The Vanuatu trench and arc mark a segment of the convergence zone between the two major plates of the Southwest Pacific region (Australia and Pacific plates). The junction area around 22°S is very active tectonically (Monzier et al., 1984). The plunging Loyalty Ridge supported by the Australia Plate enters and partially clogs the trench. Considering the geometry of the Loyalty Ridge, the strike of the trench and the current orientation and rate of convergence (12 cm/y on ENE-WSW), the subduction/collision of the ridge tends to increase and would have started around 0.3 Ma (Monzier et al., 1990). The data obtained by multibeam mapping and submersible diving (Daniel et al., 1986; Monzier et al., 1989 and 1990) at the junction zone (between 21.5°S and 22.2°S) indicate: 1) a spectacular collapse of the ridge as it approaches the trench (reef limestones affected by normal faulting are at 4,300 m depth), 2) a migration of the deformation front on the outer wall of the trench with the unusual presence of folds, 3) a narrowing and an eastward retreat of the trench by around 20 km relatively to its supposed initial position, 4) an uplift of the inner wall and 5) the development of E-W trending scarps suggesting left-lateral strike-slip motion. The rapid variation of the convergence vector and the presence of numerous left-lateral strike-slip faulting earthquakes around 22°S, at the front of the junction zone and along or at the rear of the Matthew-Hunter arc segment, also suggest that the subduction/collision of the Loyalty Ridge causes the development of a new left-lateral plate boundary through the overlapping plate, connecting the trench to the spreading center of the North Fiji basin and thus isolating a microplate (the Matthew-Hunter microplate) at the southern end of the arc (Louat and Pelletier, 1989). The rate of motion on this transform fault zone was estimated by these authors at 10.5 cm/year. However, its precise geometry and location are not known, and several variants have been proposed (Louat and Pelletier, 1989; Maillet et al., 1989; Monzier, 1993; Patriat et al., 2015). As these authors have partially indicated, it is likely that this left-lateral shear zone is complex and that a bookshelf tectonics occurs at the southernmost part of the Vanuatu trench (21°S-23°S), by associating with the main sinistral motion, dextral and extensive movements along NW-SE trending faults and pull-apart basins.

Series of GPS geodetic measurements on the Loyalty Ridge (Walpole, Maré, Lifou) and the Vanuatu Arc (Matthew, Hunter, Aneityum, Tanna) sites from 1992 to 2000 have confirmed the presence of the left-lateral transform fault zone (Pelletier et al., 1998; Calmant et al., 2003). The data indicate that the convergence rate (Australia fixed) of 120 mm/year at N248° north of the ridge-arc junction (Tanna, Aneityum) is partitioned toward the south into a convergence rate of 50 mm/year perpendicular (N197°) to the trench (Matthew) and a

sinistral movement of 90 mm/year along an E-W trending transform zone, cross-cutting the arc around 22°S and thus isolating the Matthew-Hunter microplate at the southern end of the arc (Calmant et al., 2003). In addition, GPS derived vectors of the New Caledonia sites are in good agreement with the movement of the Australian plate, suggesting therefore no significant intra-plate deformation between islands of the New Caledonian Archipelago. The termination of the southern Vanuatu back arc basins north of the junction zone, the increase in

seismic activity and the shift towards the trench of the seismogenic zone in front of the junction zone, the short length of the Wadati-Benioff plane south of Aneityum (less than 200 km), the weak development of the volcanic arc at the front of the junction zone, the particular chemistry of the volcanism of the termination of the arc south of the ridge-arc junction (calco-alkaline magnesian and boninitic series) as well as the offset of the central spreading axis in the North Fiji basin have also been linked to the subduction/collision of the Loyalty Ridge

(Monzier et al., 1984, 1990; Louat and Pelletier, 1989; Maillet et al., 1989; Monzier, 1993).

**2.2 Seismicity at the Loyalty Ridge-Vanuatu Arc junction**

The Loyalty Islands region and especially the Loyalty Ridge-Vanuatu Arc junction area around 22°S, 169.5°E is seismically very active. Nine large shallow earthquakes with magnitude equal or greater than seven occurred in this junction area since 1900. The largest was a M7.9 in August 9, 1901, located at 22°S, 170°E. A M7.6

earthquake occurred in March 16, 1928 at 170.24°E, 22.45°S. The other seven occurred during seismic crises in the last 40 years: a $M_w$7.4 in October 25, 1980; a $M_w$7.7 in May 16, 1995; a $M_w$7.3 in December 27, 2003; a $M_w$7.1 in January 03, 2004; a $M_w$7.0 in November 29, 2017; a $M_w$7.1 in August 29, 2018 and a $M_w$7.5 in December 15, 2018. Among these seven M7+ events, four have occurred to the west of the trench, as the result of shallow normal faulting within the Australia downgoing plate, including the two largest 7.7 and 7.5 events.

All earthquakes occurring during the crises and the period 1976-2020 and having a focal mechanism solution (CMTS) have been plotted on Figure 2a.

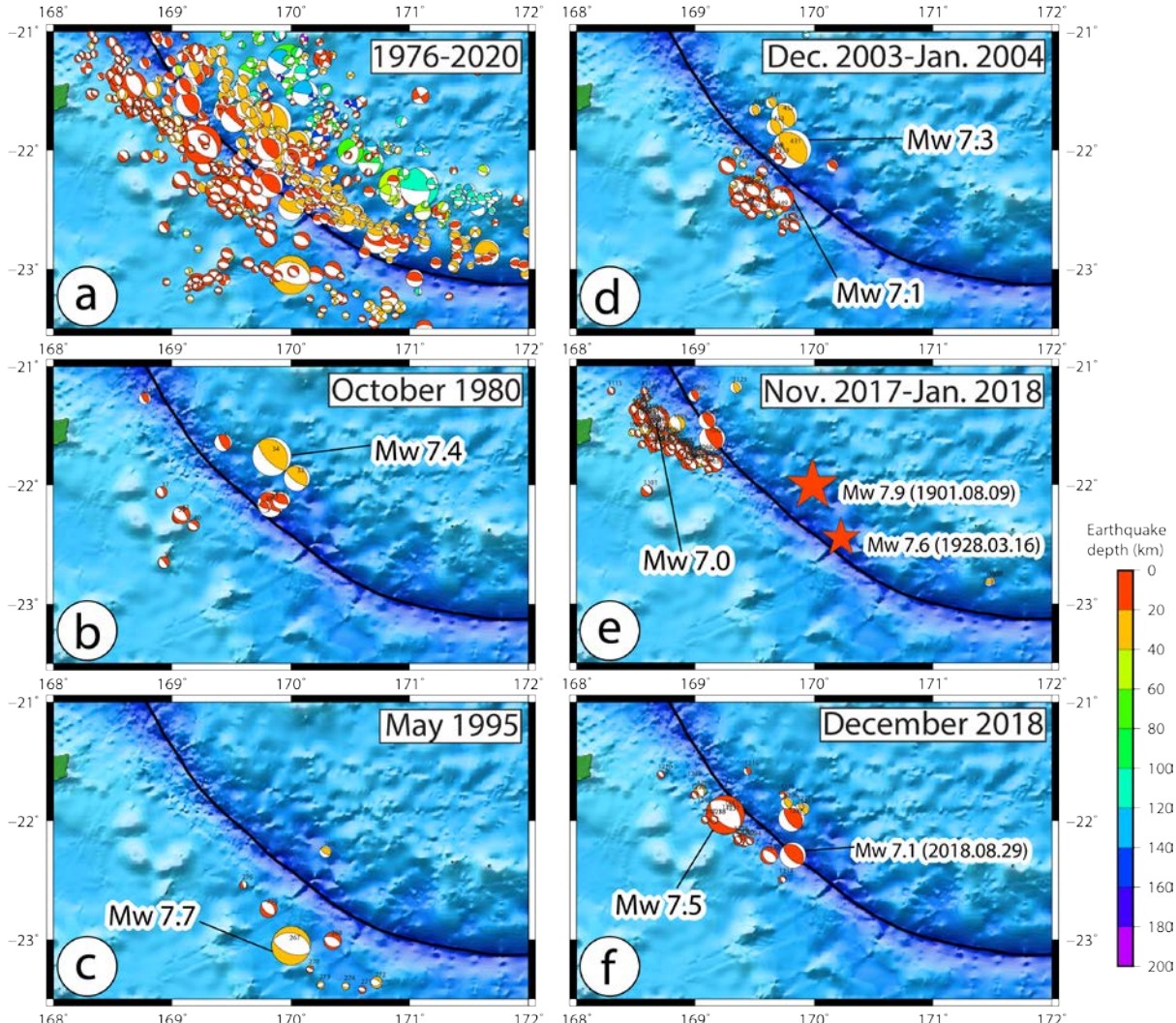

**Figure 2: Focal mechanism solutions from GCMT project plotted for the period 1976-2020 with focus on 5 different seismic crises showing 9 large shallow earthquakes at the Loyalty Ridge-Vanuatu Arc subduction zone.**

In October 1980 more than 100 events have been recorded by the worldwide network (Vidale and Kanamori, 1983). The sequence includes twelve M5.4+ events (Figure 2b). Six of them including the $M_w$7.4 main shock and two $M_w$6.5+ foreshocks are thrust faulting earthquakes east of the plate boundary and five of them are normal faulting earthquakes in the downgoing plate west of the trench. The sequence started with the three main thrust fault events and continued with the alternance of normal and thrust fault events.

During the May 1995 seismic crisis 13 events with magnitude greater than 5 were located around 23°S, 170°E (Figure 2c). Most of them are normal faulting type southwest of the trench including the $M_w$ 7.7 main shock, 125 km to the southeast of the December 2018 event. This $M_w$7.7 event is one of the largest normal faulting earthquakes known in the World in a plunging plate on the trench outer slope (Rouland et al., 1995). In detail, this earthquake and its associated aftershocks are located at the foot of the Loyalty Ridge in the adjacent South

Fiji Basin. These normal type events affecting the crust of the South Fiji Basin (from 169.75°E to 171°E) are farther from the axis of the trench relatively to the normal faulting events of the December 2003 and 2018 sequences which are on the Loyalty Ridge (169.5°E). This difference could be explained by a different rheological behavior (more buoyancy of the ridge).

Between December 25, 2003 and January 5, 2004, a shallow seismic swarm very similar to the one of 1980 occurred (same zone, same magnitude and same spatial organization of fault types; Figure 2d) (Régnier et al., 2004). More than 1000 events were recorded by the local IRD seismic network, among which about 270 by the worldwide network. Those include 37 events with magnitude greater than 4.9, 12 with magnitude equal or greater than 6 and two greater than 7. The sequence started with normal faulting events with magnitude up to 6.8 west of the trench, continued with several interplate thrust faulting events including the large $M_w7.3$ event on December 27 and located immediately to the east of the trench, and ended by normal faulting events including a large $M_w7.1$ event on January 3 located again southwest of the trench.

An important seismic crisis occurred from November 2017 to January 2018 with several thousands of events located about 70 km-100 km northwest of the December 2018 swarm (Figure 2e). Among them, 350 M4+ events have been recorded and most of the 80 M4.7+ events are normal faulting earthquakes located west of the trench along the northern edge of the Loyalty Ridge. However, in detail, the sequence began by a $M_w6.7$ and then a $M_w5.9$ thrust faulting earthquakes on October 31, 2017 and continued with numerous normal faulting foreshocks and the $M_w7.0$ normal faulting main shock on November 19, 2017.

The December 5, 2018 $M_w7.5$ earthquake can be considered as part of a seismic crisis that began on August 29, 2018 with a $M_w7.1$ interplate thrust faulting earthquake located southeastward (Figure 2f). The $M_w7.5$ normal faulting main event located west of the trench was preceded 4 minutes. before by a $M_w6.3$ event (magnitude estimated as 5.8 by the local ORSNET network) and more interestingly was followed 2h25 later by a $M_w6.8$ interplate thrust faulting east to the trench. During December 2018, about 89, 49 and 18 aftershocks of M 4+, M4.5 and M5+ respectively have been recorded by the local network.

It appears clearly that the successive seismic crises are quite similar and included both interplate thrust fault type earthquakes northeast of the trench and normal fault type events southwest of the trench in the plunging plate (Figure 2). The strong spatiotemporal pattern between these two types of events suggests that static stress interactions may account for triggering non-distant earthquake, normal faulting on the plunging plate leading to interplate thrust faulting or the reverse.

## 3 The December 5, 2018 tsunami

The tsunami following the December 5, 2018 $M_w7.5$ earthquake has been recorded by tide gauges in New Caledonia and Vanuatu but also at a regional scale (Figure 3). In addition, several observations of tsunami waves in locations not equipped with sensors provided important information to consider in the study of the event (Figures 3 and 4).

### 3.1 Tide gauge records

The tide gauge of Maré Island, located in Tadine's Harbor on the southwest coast of the island, was the first to record the tsunami signal at 4:43 UTC (3:43 PM local time – UTC+11), 25 minutes after the main shock even if local people reported an earliest arrival at the southeasternmost village (Kurin) of this island ~15 minutes after the earthquake. Then, the wave train reached the other tide gauges located in New Caledonia (4:43 UTC in Wé, Lifou Is.; 5:11 UTC in Ouinné; 5:10 UTC in Thio; 5:27 UTC in Hienghène) as well as several pressure gauges

like in Poindimié, east coast of New Caledonia. According to Roger et al. (2019b) for what concerns New Caledonia only, a maximum tsunami height of ~60-70 cm was recorded by Ouinné tide gauge.

In Vanuatu, it reached the tide gauge located at Lenakel Harbor (Tanna Island) at 4:41 UTC, showing a maximum height of ~1.5 m (amplitude of ~75 cm a.s.l.). In Efate (5:06 UTC in Port Vila), the tsunami has been also recorded on the tide gauge located at Port Vila where it reached a maximum height of ~50 cm (maximum

amplitude of ~25 cm a.s.l.).

The tsunami was also recorded by tide gauges in other locations of the southwestern Pacific region, as far as Port Kembla, Australia, about 2200 km away from the source, North Cape, New Zealand (~1400 km southward), or Pago Pago in the American Samoa's (more than 2250 km northeastward). As far as known, except in New Caledonia and Vanuatu, it never reached more than 30 cm high. Figure 3 locates the different tide gauges that

were able to record the tsunami within the southwestern Pacific Region and illustrates the recorded maximum wave amplitudes obtained after de-tiding the time series coming from different origins.

For Maré, Ouinné, Thio and Hienghène, the data comes directly from the raw dataset of the pressure sensors. For Lifou, the data have been provided by the SHOM (http://refmar.shom.fr/en/lifou). The data used for Lenakel and Port Vila as well as places outside New Caledonia and Vanuatu comes from the IOC database (http://www.ioc-

sealevelmonitoring.org/) and the data for Poindimié comes from a local New Caledonia coastal water monitoring project (ReefTEMPS project: http://www.reeftemps.science/en/data/, Varillon et al., 2018).

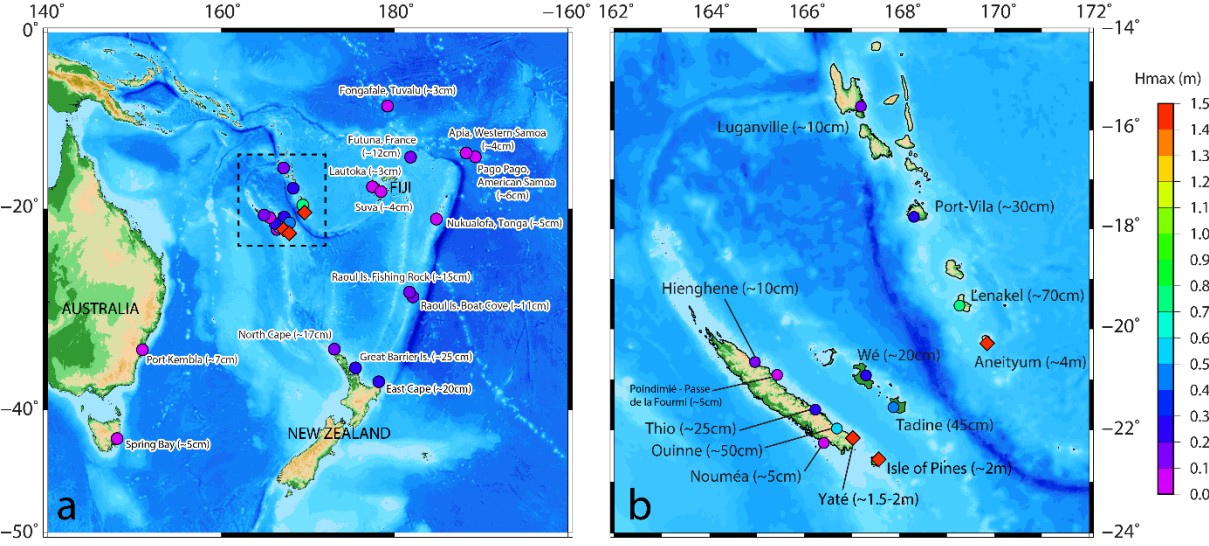

**Figure 3: Tsunami maximum wave amplitude of the December 5, 2018 tsunami in the southwestern Pacific Region. Monitored amplitudes are represented with colored circles while those coming from witness observations are represented by diamonds.**

*Note about the tide*

The arrival time in Tadine Harbor corresponds to 48 minutes before high tide (high tide at 4:30 PM local time)
and about one or two minutes after high tide in Hienghène (Northeast of Grande Terre, New Caledonia) where high tide was at 4:25 PM local time and the tsunami arrival was recorded at 4:26-4:27 PM local time.

**3.2 Eye-witnesses' observations**

In the aftermath of this event, two videos showing the tsunami arrival at two different locations have been collected.

The first video from Yaté (Figure 4a), southeast coast of Grande Terre, circulating on social networks the day of the event, is very informative. It shows the arrival of the tsunami over the fringing reef shelf exposed out of the water by a first important withdrawal of the sea between ~100 and 200 m; note that the sea was reaching nearly high tide at the moment of the tsunami arrival with a predicted maximum water level of 1.55 m at 4:31 PM local time at Ouinné, the nearby tide gauge, corresponding to a water depth of ~1.2-1.3 m over the reef shelf in Yaté.

Two quantitative pieces of information come from the analysis of the video. The first one is an estimate of the tsunami speed from ~ 5 to 10 $m.s^{-1}$ (18 to 36 $km.h^{-1}$). The second one is the maximum tsunami height of ~2.3 m reached in ~20 s (after the withdrawal), derived using one isolated mangrove tree exploited as a flood scale afterwards (Figure 4E).

The second video and related pictures show the tsunami arrival and its consequences at Le Méridien Resort

(Figure 4b), Isle of Pines, the southernmost island of New Caledonia, and have been provided courtesy of M. Bretault (Technical Director of Le Méridien Resort of the Isle of Pines). The video shows the tsunami travelling into the shallow channel that encircles the resort complex and its surrounding (Figure 4B1). With the help of aerial orthophotos (Government of New Caledonia, tile n°55-17-IV, https://georep-dtsi-sgt.opendata.arcgis.com/pages/orthophotographies), one can derive the tsunami speed in the channel of around 5

$m.s^{-1}$ (18 $km.h^{-1}$). The pictures have been taken after the tsunami and reveal the damages on several bungalows and around the swimming pool, and show the run-up extent of the waves (Figure 4B2).

In Vanuatu, the tsunami has been reported in several places from Aneityum Island in the south, to Tanna, and Efate Islands. It reached Aneityum first, where the impact has probably been the worse in the wholeregion affected by the tsunami; the effects were severe especially in Umetch area where it washed the coast and the

plantations with waves reaching ~4 m (Tari and Siba, 2018) and penetrating more than 200 m inland (Vanuatu Daily Post, December 6, 2018) as shown on Figure 4c, leaving people homeless. It has also badly damaged Mystery Island and its airport on the southwest of Aneityum, a major source of incomes for the island. Other places like Anelghowhat have also experienced the tsunami but without important damage as reported in the Vanuatu Daily Post (December 8, 2018). Then it reached Tanna where it has been recorded by Lenakel tide

gauge as reported hereabove but it has also been reported by the manager of Ipikel, a village on the southeastern coast of the island (Sulphur Bay), as having reached the first houses without any damage, about 80 m from the shoreline and ~1.5 m high (Isaac, manager of Ipikel, pers. comm., 2019). In Efate, witnesses reported a small inundation on Erakor Island, south of Port Vila (Figure 4d).

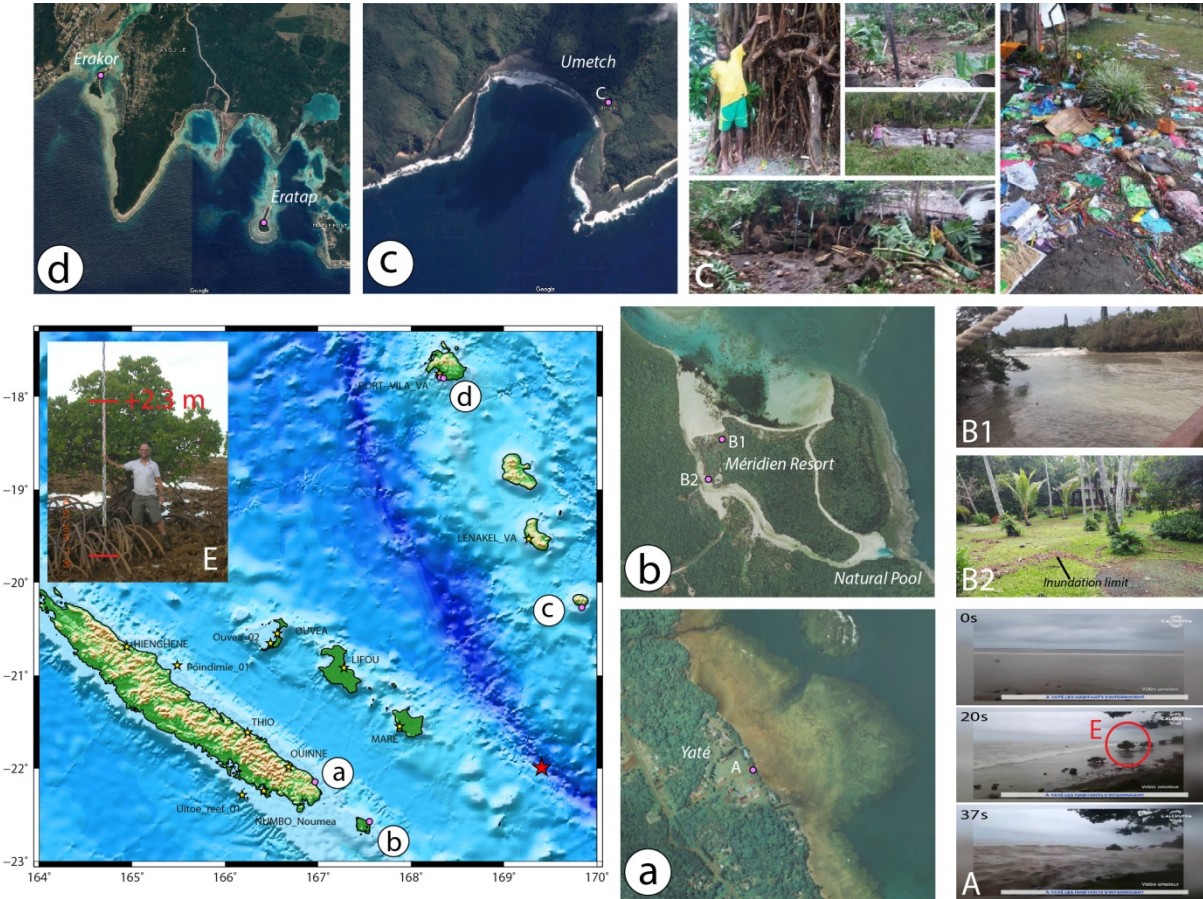

**Figure 4: Observations of the tsunami arrival and height in several places in New Caledonia and Vanuatu. a: Yaté; b: Pine Island-Le Méridien Resort; c: south Aneityum, d: south of Port Vila, south Efate. (Photos credit: a, b: © Georep New Caledonia 2021; c, d: © Google Earth 2021 - CNES/Airbus; A: Caledonia TV; B1 & B2: Moana Bretault; C: Vanuatu Meteorology and Geohazards Department; E: authors).**

## 4 Tsunami numerical simulations

In this section, details about the Digital Elevation Model (DEM) used in computational grid generation, models used to simulate bottom deformation, tsunami generation and propagation and sensitivity tests to detail of the rupture model are presented.

### 4.1 Input data

### 4.1.1 Bathymetric grids

It is well known that tsunami's behavior is dependent upon the bathymetric features and the coastal geometries (e.g., Matsuyama, 1999; Hentry et al., 2010; Yoon et al., 2014). When it approaches coastlines or seamounts, the wave shoaling leads to the rising-up of the amplitude and slows down the tsunami as the water depth reduces. It is even worse when the tsunami enters harbors, bays, lagoons or fjords able to produce resonance, a phenomenon particularly well studied during the two last decades (e.g., Barua et al., 2006; Rabinovich, 2009; Roger et al., 2010; Roeber et al., 2010; Bellotti et al., 2012; Vela et al., 2014; Aranguiz et al., 2019). It is also possible that a

resonant behavior occurs between neighboring islands like it happened in Hawaii during the 2006 Kuril tsunami (Munger and Cheung, 2008).

For these reasons, it is necessary to model tsunami propagation on bathymetric grids keeping the most relevant details. There are two main traditional downscaling strategies in wave and tsunami modelling. One uses a sequence of nested structured-grid models; the other relies on a single unstructured-grid model. Both techniques aim at obtaining high-resolution wave fields in shallow area and provide similar results (Harig et al., 2008; Pallares et al., 2017), even though several studies have highlighted that the use of only one unstructured mesh grid for tsunami modelling provides better reproduction of tsunami observations and records in comparison to nested grids scheme use (e.g. Harig et al., 2008; Shigihara and Fujima, 2012). When considering the presence of many archipelagos forming the Melanesian volcanic arc (Solomon Islands and Vanuatu, Figure 3) and peculiar details along the New-Caledonia's coastline (Figure 4), the unstructured grid method provides multiple advantages. This technique allows more flexibility in mesh design and can capture more coastline details than regular meshes at the same computational cost.

The bathymetric grid is an unstructured mesh forming a triangular irregular network (TIN) DEM (Figure 5) with varying mesh size (from 5 m along the coastline to 2150 m in the deep ocean, with a median value of 70 m, corresponding to the target size for grid resolution along the coastline), used for calculation of tsunami generation, propagation and interaction with the shallow water features.

The TIN DEM generation has been made with Shingle 2.0 (Candy and Pietrzak, 2018), an automatic grid generation algorithm, using: 1) Smith and Sandwell (1997) v. 8.2 dataset, 2) an extended ~180 m resolution DEM covering the whole economic zone of New Caledonia and Vanuatu, based on single and multibeam echosounder data and produced especially for the assessment of tsunami hazard in New Caledonia and 3) 10 m resolution data on harbors where tide gauges and/or witnesses' observations are located. This latest data comes from digitized nautical charts, aerial pictures interpretation and multibeam bathymetric surveys.

A variable mesh size function is designed to capture the evolution of the tsunami wave with a spatial discretization of 30 points per wavelength. Along the coastline or places with shallow features and gauge stations, additional mesh refinement rules are imposed in the mesh size function. Figure 5b, 5c and 5d illustrate the increase of spatial resolution when approaching the barrier reef and the coastline.

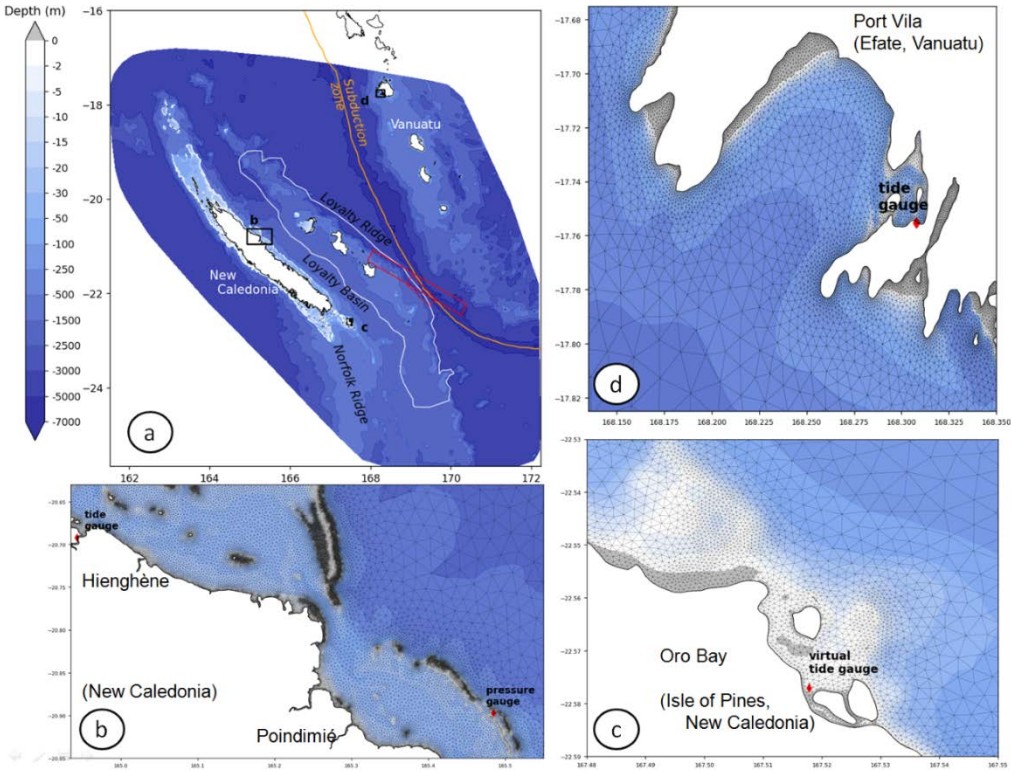

**Figure 5: Triangular irregular network (TIN) DEM including New Caledonia and South Vanuatu Islands. The three insets show mesh details and location of some gauges (see black boxes b, c and d for exact location). In a), extents of both the non-uniform and uniform slip models considered in this study are displayed with solid and dashed red lines respectively. The white contour indicates the area of the Loyalty Ridge where the ocean bottom is modified to further investigate the impact of this ridge on the wave propagation (see Discussion and Figure 10).**

### 4.1.2 Initial deformation

The location of the March 5, 2018 earthquake and the different focal mechanism solutions indicate that the earthquake was the result of shallow normal faulting along a fault plane trending NW-SE within the plunging Australia Plate on the northern border of the Loyalty Ridge. USGS, GCMT and GEOSCOPE (SCARDEC) propose a magnitude $M_w$ 7.5 to 7.6 and parameters for the rupture (strike, dip, rake) of respectively [298°, 43°, -111°], [312°, 36°, -90°] and [297°, 55°, -108°].

Analysis of seismic data by USGS indicates a rupture duration of about 50 s and 3 phases of displacement during the rupture (https://earthquake.usgs.gov/earthquakes/eventpage/us1000i2gt/finite-fault). Using inversion in the wavelet domain of teleseismic broadband data and long period surface waves (Ji et al., 2002), USGS proposes a non-uniform fault model (called NUM hereafter) of 272 km x 40 km composed of 272 fault segments of 8 km x 5 km with a slip ranging from a few millimeters up to 3 m mainly distributed in the 15 km upper part of the fault plane (hypocenter being at 12 km) and a maximum displacement patch at an along strike distance around 40 km northward of the hypocenter. All segments have the same orientation parameters for azimuth (298°) and dip (43°).

Considering the variability of parameter values, the geological and tectonic context and the effects of the tsunami along the shores of New Caledonia, as well as the role played by submarine features in the tsunami propagation, sensitivity tests have been computed through uniform slip rupture scenarios to assess the importance of each

parameter on the tsunami amplitude at key locations. A uniform slip model scenario (called UM hereafter) based on the GCMT solution has been built choosing its parameters in agreement with the geological knowledge of the region and the empirical relationships from Strasser et al. (2010) and Blaser et al. (2010). These relationships link the earthquake seismic moment magnitude to the geometry of the fault plane considering the faulting condition (interslab or intraslab). It helps to estimate a rupture length $L$ of 80 km and a rupture width $W$ of 30 km (i.e. a rupture surface $A$ of 2400 km²) corresponding to the seismic moment $M_o$ of 2.52 e+20 N-m estimated by GCMT. The relationship $s = \frac{M_0}{\mu A}$ gives the coseismic slip on the fault plane $s = 3.5$ m with a rigidity (or shear) modulus $\mu$ of 3 x $10^{11}$ dyne cm$^{-2}$.

Each seafloor initial deformation has been computed using the Okada's (1985) static surface deformation analytical expressions modelling an earthquake rupture in an elastic half-space. The shape of this initial deformation is directly linked to essential parameters like the depth of the hypocenter and the movement on the fault plane. This initial deformation is transmitted to the water surface above for further tsunami simulations assuming an incompressible fluid.

Table 1 details fault geometries and orientation parameters (Φ: strike, δ: dip and λ: rake) for NUM and UM.

**Table 1 : Rupture parameters used for each scenario NUM and UM.**

| Scenario | Length | Width | Moment | Depth | Slip distribution | Strike | Dip | Rake |
|---|---|---|---|---|---|---|---|---|
| NUM | 272 km | 40 km | 2.49 e+20 N-m | 4 to 28 km | 0 to 3.0 m | 298° | 43° | -80° to -150° |
| UM | 80 km | 20 km | 2.52 e+20 N-m | 17 km | 3.6 m | 312° | 36° | -90° |

As there is a large body of evidence that small changes in Φ, δ and λ angles may impact both the wave propagation over rugged seafloor (for example see Burbidge et al. 2015) and amplitudes of leading tsunami waves in exposed points (Necmioglŭ and Özel, 2014), sensitivity tests have been also conducted using different sets of fault orientation parameters. As detailed in table 2, we use variations in Φ/ δ /λ angles from values used in the UM case. The range of tested values for Φ/δ /λ is based on the geological knowledge of the region's faults.

**Table 2 : Rupture orientation parameters investigated in sensitivity tests**

| Strike (Φ, degrees) | Dip (δ, degrees) | Rake (λ, degrees) |
|---|---|---|
| 290.0/298.0/305.0/312.0/320.0 | 25.0/36.0/43.0/55.0/60.0 | -120.0/-110.0/-100.0/-90.0/-80.0 |

## 4.2 Tsunami generation and propagation

Tsunami waves generated by the seafloor displacement and their propagation are computed using the Semi-implicit Cross-scale Hydroscience Integrated System Model (SCHISM), an unstructured ocean model developed by the Virginia Institute of Marine Science (Zhang et al. 2015, 2016a) based on the former 3D ocean model SELFE from Zhang and Baptista (2008). It is an open-source community-supported ocean model heavily tested and under continuous improvement in laboratories worldwide, oriented towards a handful of different modelling

domains using specific modules like wind-wave modelling (e.g. Roland et al., 2012; Hsiao et al., 2020), sediment transport modelling (e.g. Pinto et al., 2012; Lopez and Baptista, 2017) or tsunami modelling (e.g. Zhang et al., 2016b; Priest and Allan, 2019). Modelling of tsunami propagation and coastal interaction is performed through unstructured grids like TIN. Inundation could also be calculated but the authors have decided not to do it due to the inadequacy of topographic data. According to Horrillo et al. (2015), SCHISM has passed successfully the United States of America NTHMP (National Tsunami Hazard Mitigation Program) benchmarks from the OAR-PMEL-135 standard providing a list of exercises like the famous 1993 Okushiri tsunami exercise (https://nctr.pmel.noaa.gov/benchmark/index.html).

SCHISM is capable of solving the 3-D Reynolds-Averaged Navier-Stokes (RANS) equations. It uses a semi-implicit Galerkin finite-element and finite-volume method on unstructured grids (Zhang and Baptista, 2008; Zhang et al., 2016a, 2016b) with time stepping with no CFL (Courant-Friedrich-Lewy) stability/convergence condition. This way, large time steps could be applied even with high resolution meshes. In this study, SCHISM is used in barotropic mode with hydrostatic assumption and one layer. In 2-D mode, RANS equations are depth-integrated, and the circulation is described using Non-linear Shallow-water Wave equations (NSW), a simplification widely used to model tsunamis. Neglecting wind stress, earth tidal potential and atmospheric pressure forces, the NSW equations used in SCHISM 2-D at point (x,y) with depth h below the geoid are :

Continuity equation: $\frac{\partial(\eta\text{-}b)}{\partial t} + \nabla.(\mathrm{u}H) = 0$

Momentum equation: $\frac{\partial \mathrm{u}}{\partial t} + (\mathrm{u}.\nabla)\mathrm{u} = f(v,-u) - g\nabla\eta - f_{hd} - \frac{\tau_b}{H}$

Here, $t$ is time, u(x,y,t) the depth averaged horizontal velocity with components ($u,v$), $\eta$ the sea surface elevation above the geoid, $b$ the time-dependent seabed displacement (positive for uplift), H the total water depth (H=$\eta$-$b$+$h$), $f$ the Coriolis factor, $g$ the gravity acceleration, $f_{hd}$ the horizontal eddy viscosity (set to $10^{-4}$ m$^2$.s$^{-1}$) and $\tau_b$ the bottom drag following a quadratic form:

$$\tau_b = g\frac{M_n^2}{H^{1/3}}\|\mathrm{u}\|\mathrm{u}$$

where $M_n$ is Manning's roughness coefficient set spatially uniform with a value of 0.025 s.m$^{-1/3}$. All tsunami simulations were performed assuming that prevailing tide was static (no flow) and equal to high water (+1.6m). To limit undesirable wave reflection, a Flather radiation condition (Flather, 1987) is applied along the open boundaries with specified outer values 0 m.s$^{-1}$ and 1.6 m for U and $\eta$ respectively.

In a first step, SCHISM is used to generate the sea-surface initial deformation and flow dynamics in response to the bottom motion. The dynamic displacement of the seafloor can be described in SCHISM by adding a time dependent seafloor displacement term $b$ incorporated in NSW governing equations. This is done by multiplying Okada's static solution $b_0$ by a uniform rate function of the rising time. In agreement with seismic records, a rising time of 50 s has been used and SCHISM was run with a time stepping dt = 1 s. During the rising time, the seafloor anomaly $b_0$ is progressively injected to give the initial condition for the free surface and horizontal momentum conditions. Then, to simulate tsunami propagation, the model runs with dt = 30 s for a duration of 3

380  hours. It is worth noting that using the default value of 10 s for the rising time, leads to marginal effects on results.

To detect changes due to fault parameters, total wave energy ($E$, unit j.m$^{-2}$) is added in SCHISM outputs, as the sum of two components, kinetic energy (first term) and gravitational potential energy (second term):

$$E = \frac{1}{2}\rho H U^2 + \frac{1}{2}\rho g \eta^2$$

It is important to underline that the sea-level has been set to a high tide value of 1.6 m, which was approximately
the situation in most places when the tsunami reached New Caledonia and Vanuatu on December 5, 2018.

**5 Simulation results**

**5.1 Waves energy**

Figure 6 presents the maximum wave energy maps obtained after 3 hours of tsunami propagation over the TIN DEM for NUM and UM. The first observation is that NUM is accompanied by much less tsunami wave energy
than UM. Within the two tsunami beams propagating from the rupture location, there are differences in wave energy higher than 10% in the deep ocean. But, in shallow areas, like banks and seamounts near the rupture, there is a 100% change. In exposed locations, like Isle of Pines and Aneityum in the SW and NE quadrants, respectively, wave energy anomalies are higher than 50%, implying lower simulated wave amplitudes in those locations using NUM instead of UM. It is also very striking that despite its proximity and facing the NUM
rupture fault, simulated wave energy along the western coast of Maré is lower compared to UM.

The second observation is that, in both cases, the maximum wave energy field is mainly oriented in the direction perpendicular to the azimuth of the fault, i.e. NE-SW, with respect to the slip angle (=rake) (Okal, 1988). Even if it is less obvious for NUM than UM, the wave energy is clearly captured by the Loyalty Ridge, which supports the Loyalty Islands, and the Norfolk Ridge which is the extension of the Grande Terre of New Caledonia
towards the south. This refocusing of the wave train in another direction is due to the fact that the tsunami speed relies only on the bathymetric depth in the open ocean (Satake, 1988; Titov et al., 2005; Swapna and Srivastava, 2014). Thus, if the waves encounter submarine features like seamounts or ridges, which means that the sea depth decreases, the trajectory of the tsunami could be considerably modified. In the present case, the Loyalty and Norfolk Ridges acting like waveguides help the waves to propagate in the azimuthal direction toward the
northwest (Loyalty Islands and Grande Terre).

The wave energy difference between the two models shown on figure 6 highlights that the main coastal differences concern the Isle of Pines, Maré and Aneityum within a range of 20 to 60% more energy for UM than NUM. Along the east coast of the Isle of Pines, the energy increase is in the range 20% to 30% and up to 50% near specific coastal features like bay entrances. Along the south coast of Aneityum, the closest observation site
in the main energy path of the tsunami, the total wave energy increases by about 30%.

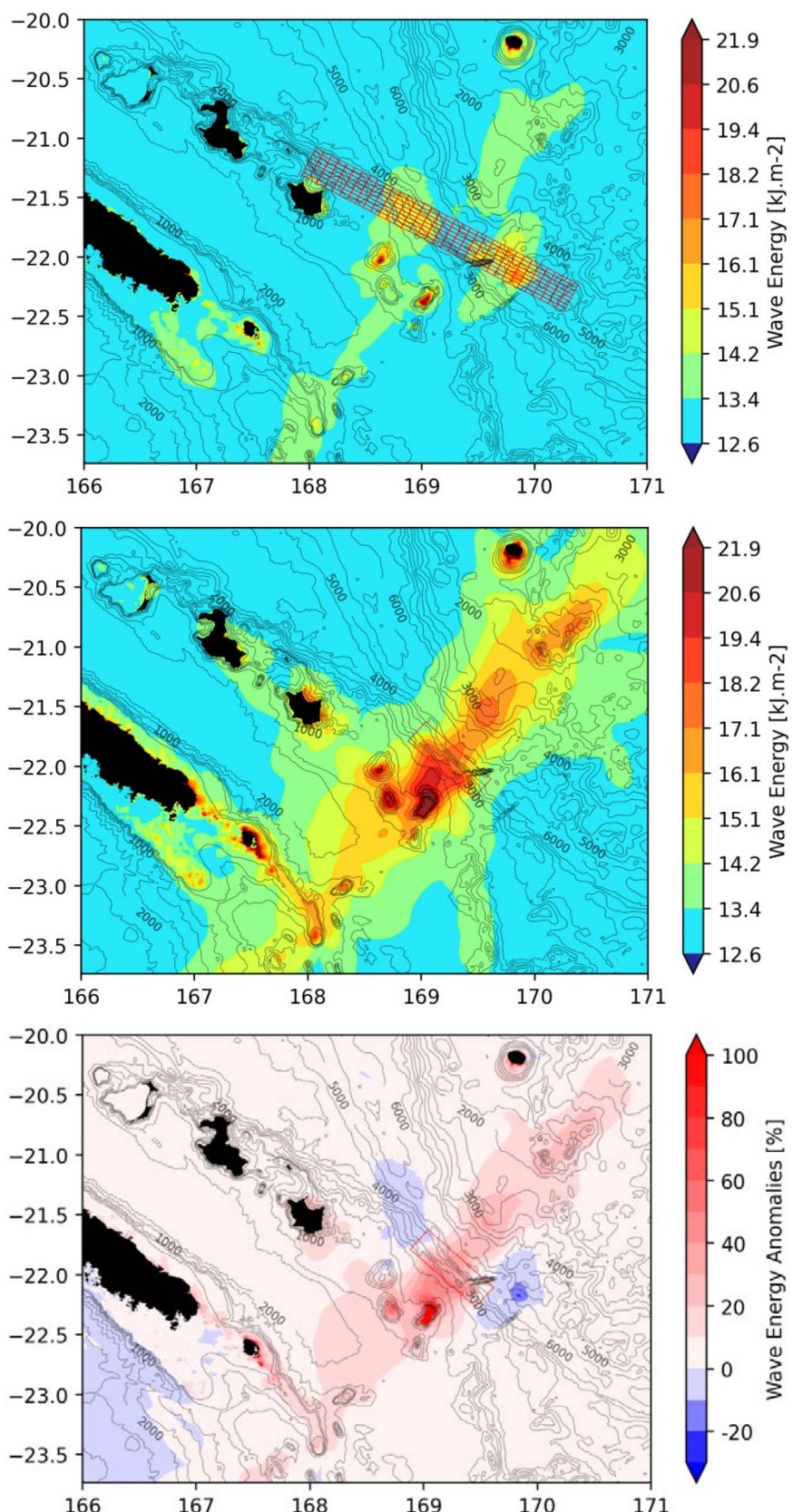

**Figure 6: Total wave energy maps for two different models: NUM with strike = 298° (top), UM with strike = 312° (middle) and relative energy anomaly between the two (bottom). The bathymetric contours underline the features playing a possible role in tsunami wave propagation. The extent of the sources is symbolized with the red rectangles: small boxes (top) denote the 272 fault segments solution from USGS where heterogeneous slip distribution (NUM) is applied along a 270km long rupture fault. In the UM case (middle), uniform slip deformation is applied over the 80km long rupture model.**

### 5.2 Tide gauges' records

Simulated time series of sea level variation from the two scenarios are compared to maregraphic records onfigure 7. Clearly, at all stations, NUM does not fit well the recorded amplitudes: the simulated amplitudes of the first leading waves are very low in comparison to records. In terms of arrival times, it is globally in good agreement with the UM scenario except for Maré, as a consequence of the exaggerated extension of the rupture fault toward this island provided by USGS. There is some evidence that the heterogeneous slip distribution and geometry from USGS is not appropriated and a simple model for rupture like UM is still more justified for that event. In this basis, further investigations will continue with UM leaving aside the NUM scenario.

Synthetic time series obtained with UM show that,

- at Tadine (Maré), the modelling is not able to reproduce correctly the tide gauge record in terms of arrival time and wave amplitude (Figure 7a). It shows a delay of ~5 min, the simulated signal arriving earlier than the observed one. Also, it does not reproduce the oscillation of period ~4-5 min with amplitudes more than three times larger than the modeled ones.
- at Wé (Lifou), the simulated signal exhibits some strong similarities with the real one recorded in terms of polarity, wave amplitude and periodicity, but there is a delay of more than 5 minutes, the modelling being faster than the reality (Figure 7b).
- at Thio, the simulated signal is able to reproduce the real record for what concerns the polarity, the amplitude or the periodicity but not exactly the arrival time, being still early of a couple of minutes (Figure 7c).
- at Ouinné, the modelling is not able to reproduce the recorded signal, except for the first wave polarity; the simulated record arrives 5 min beforehand (Figure 7d). An oscillation with a period of ~6-8 min seems to occur after the first arrival.
- at Poindimié - Passe de la Fourmi, there is a good agreement between modelling and observations: the arrival time exhibits only a small shift of 1-2 min, the modelled signal being the fastest (Figure 7e). The wave amplitude and polarity are quite good, and the periodicity shows only a few differences that will be discussed further.
- at Hienghène, the modelled tide gauge record is observed again to arrive in advance by about 2-3 min with respect to the real one (Figure 7f). The wave polarity and periodicity are well reproduced, but the amplitude is slightly overestimated by the modelling.
- in Vanuatu, at Lenakel, Tanna, there is good agreement between the arrival time and first wave amplitude of the modelled and real tsunami signal (Figure 7g). But the periodicity and amplitudes are strongly different, the modelling being unable to reproduce what looks like a resonant oscillation with a period of ~6 min and a maximum amplitude reaching nearly 40 cm around 25 min after the first tsunami wave arrival.

- at Port Vila the simulated signal well reproduces the tide gauge record in terms of arrival time ~40 min after the earthquake (exhibiting only a small delay of ~1-2 min), but also in terms of polarity, wave amplitudes and periodicity (Figure 7h). Note that the large trough and peak occurring after 100 min are not reproduced satisfactorily in the simulation.

It is worth noting that sea level records from several stations located in bays and harbors exhibit large oscillations typical of harbor resonance triggered by the first leading waves. It is worth noting that with the actual model settings for SCHISM (30 nodes per tsunami wavelength and time step dt = 30 s), the model seems unable to reproduce resonance in harbor (Wé, Tadine, Lenakel) or semi-enclosed bay like Ouinné. Since such wave amplification processes represent a significant, but undocumented threat in New Caledonia, future works will be devoted to the representation of harbors and bay resonances due to tsunami with SCHISM.

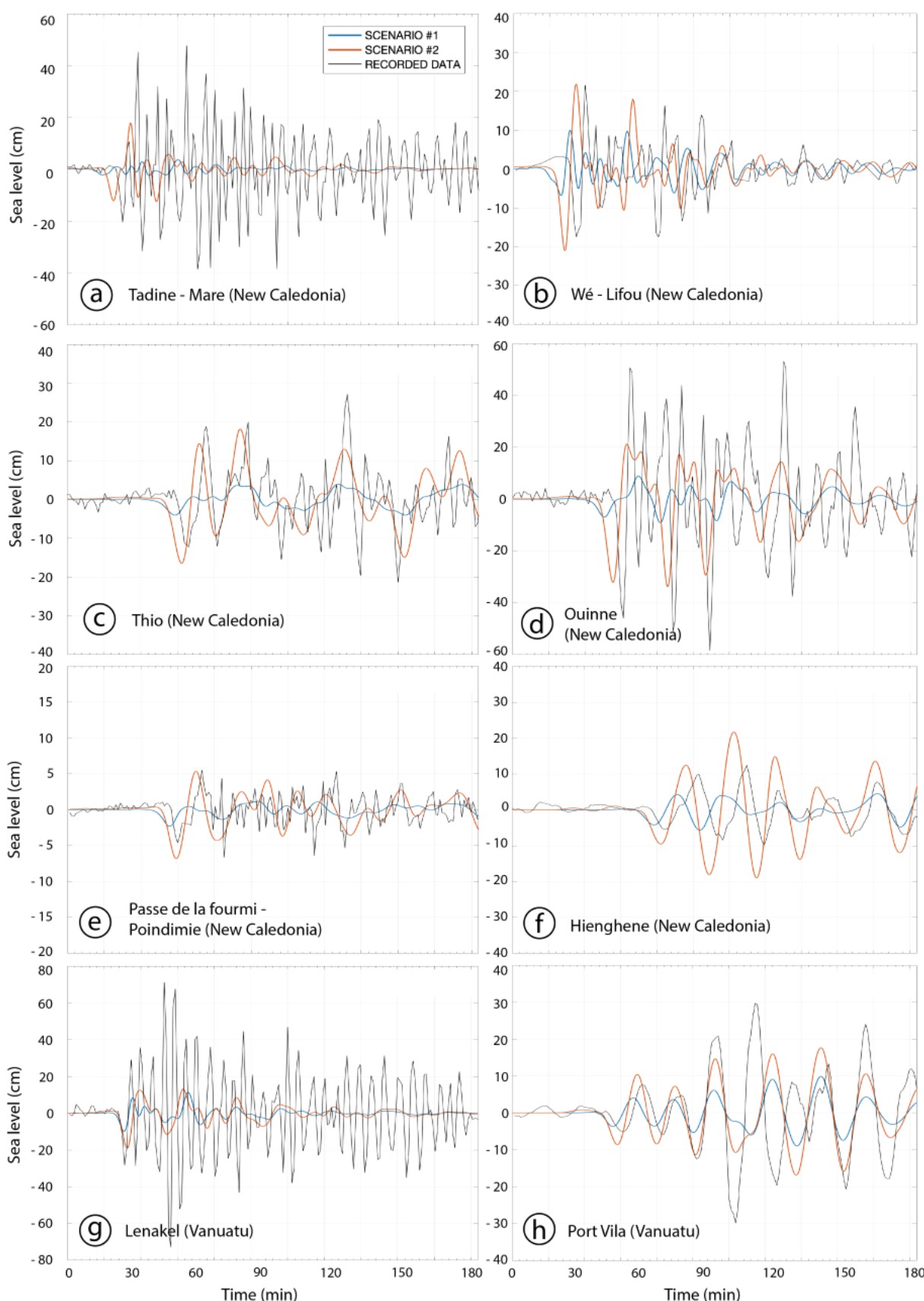

**Figure 7 : Comparison between real (black) and simulated records (blue: NUM; red: UM) for 8 different tide gauges located in New Caledonia (a, b, c, d, e, f) and Vanuatu (g, h). These tide gauges are located on figure 3b. Time is relative to the earthquake occurrence time (4:18 UTC). Be careful to the sea level scale for each figure.**

### 5.3 Tsunami maximum wave amplitude

The tsunami energy is partially captured by the submarine ridges oriented perpendicular to its main propagation way, leading to amplifications in the Loyalty Islands (via the Loyalty Ridge) and around the Isle of Pines (via a series of seamounts and guyots constituting the south-eastern seamounts complex of the Pines Ridge). The TIN DEM allows zooming onto specific areas like Aneityum (Figure 8b), the Isle of Pines (Figure 8c), Yaté (Figure 8d), Port Vila (Figure 8e) and Sulphur Bay, Tanna Island (Figure 8f) helping to further compare the testimonials

to the simulation results. Important coastal amplifications of the tsunami occur along the south coast of Aneityum from Mystery Island to Umetch, showing maximum wave amplitude of more than 1.5 m between Mystery Island and the main island (Figure 8b). Coastal amplification is also relatively important in some restricted locations along the east coast of the Isle of Pines (Figure 8c) showing wave amplitude of more than 1 m in front of the Le Méridien Resort but also ~ 40-50 cm in the bay of Ouaméo on the west coast and the Crab's

Bay in the north of the island. Wave amplification along the coast of Yaté (south-eastern part of Grande Terre, Figure 8d) leads to maximum wave amplitude of ~50 cm in front of the church of Touaourou and in the Yaté River estuary. Focus on Port Vila, located along the south coast of Efate Island (Figure 8e) and on Sulphur bay, southeast of Tanna Island (Figure 8f), show wave amplification in a few places, reaching ~40 cm maximum in both cases.

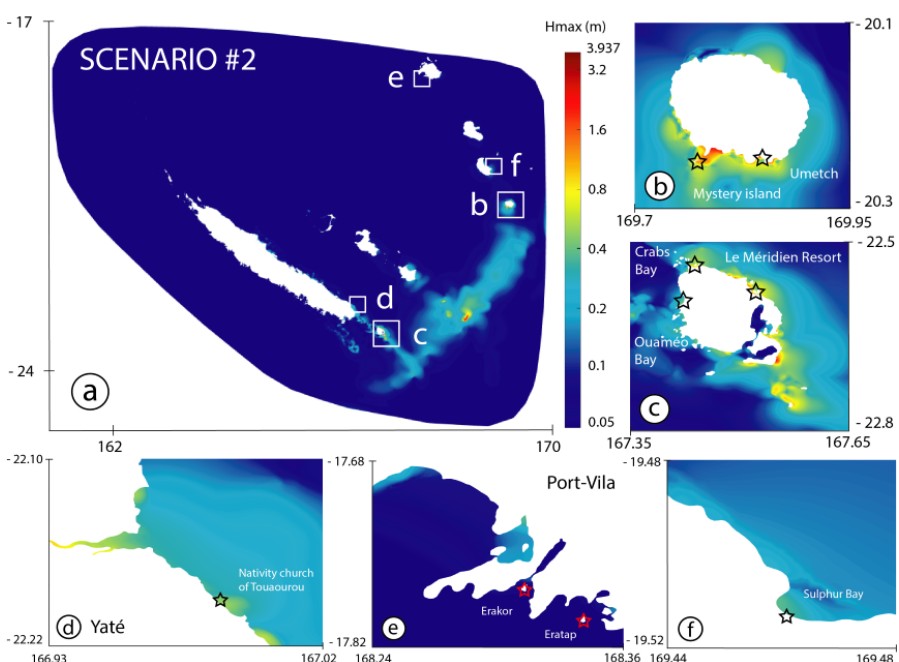

**Figure 8: Maximum wave amplitude maps ($H_{max}$) obtained after 3 hours of tsunami propagation on the TIN DEM for scenario 2 (UM) in New Caledonia and South Vanuatu. a: across the entire area, b: Aneityum island, c: Isle of Pines, d: Yaté; e: Port vila, Efate; f: Sulphur Bay, Tanna. Stars stand for eye-witnesses observation points.**

### 5.4 Sensitivity study

Table 3 and figure 9 detail model results concerning the sensitivity of uncertainties in fault angle parameters ($\Phi$: strike, $\delta$: dip and $\lambda$: rake) on the maximum generated tsunami wave amplitude ($H_{max}$) and tsunami travel time (TTT) at key locations in New Caledonia and Vanuatu. Table 3, row a., gives the range of variation for all key

locations for both $H_{max}$ as a function of Φ, δ and λ. In general, changing azimuth (Φ) of the UM source from 290° to 320° and keeping all other parameters stable, result in large variations in exposed locations like in front of Le Méridien Resort Isle of Pines (New Caledonia) or at Mystery Island (Aneityum, Vanuatu) with change of about 62 and 86% respectively. It is the same with the dip and rake to a lesser extent: variations of the dip from 25° to 60° and of the rake from -120° to -80° lead to relative variations of $H_{max}$ of about 20% and 55% respectively at the same places. It is worth noting that the location exhibiting the largest change to strike angle uncertainties (with a 100% change) is We, Lifou, aligned with the rupture fault. But, in term of strike sensitivity ($\frac{dH_{max}}{d\Phi}$), the slope computed from linear regression between $H_{max}$ and Φ (see relationships on figure 9, left panel), the sensitivity to Φ at exposed location like Isle of Pines is twice the value at We, Lifou (1.2 against 0.6 cm.degree$^{-1}$ respectively).

But uncertainties in Φ/δ/λ angles have also significant control on the arrival time of the first leading wave as investigated in table 3, row b. and in the right panel of figure 9. Results indicate that it is the dip angle δ that could exert large variation in TTT, with variations up 5 to 6 minutes at Hienghène and Port Vila, the more remote location from the rupture fault considered in the model domain. Obviously, possible uncertainties in Φ/δ and λ may explain some lags between model results and observations.

**Table 3 : Results of our sensitivity tests at keys locations using three sets of parameters acting on the rupture fault orientation. There are: strike, dip and rake with values incremented as detailed in table 2. Row a.: impact on the maximum Elevation (Hmax). Row b.: impact on the travel time (TTT).**

| | Hienghène | Poindimié | Thio | Ouinné | We, Lifou | Tadine, Maré | Yaté | Méridien Res. Pines Island | Umetch, Aneityum | Mystery Isl., Aneityum | Lenakel Tanna | Port Vila, Efate |
|---|---|---|---|---|---|---|---|---|---|---|---|---|
| Row a. Maximum change in $H_{max}$ in cm (in percent from the minimum value) | | | | | | | | | | | | |
| Strike Φ [290:320] | 0.5 (2.6) | 0.7 (13.9) | 2.8 (16.8) | 6.1 (32.1) | 18.3 (102.1) | 5.6 (46.0) | 6.9 (22.9) | 40.7 (61.8) | 39.2 (44.0) | 47 (85.8) | 2.5 (14.9) | 0.8 (4.2) |
| Dip δ [25:60] | 6.6 (49.1) | 2.0 (61.5) | 4.5 (34.3) | 4.4 (20.4) | 5.9 (32.7) | 4.0 (28.5) | 7.8 (28.6) | 16.0 (19.0) | 24.3 (27.4) | 17.8 (29.0) | 1.8 (11.8) | 9.1 (56.1) |
| Rake λ [-120:-80] | 1.2 (6.3) | 0.4 (8.3) | 2.1 (13.4) | 8.0 (41.8) | 0.8 (3.6) | 2.4 (15.3) | 5.2 (16.1) | 15.6 (17.8) | 16.8 (19.4) | 26.1 (53.5) | 1.0 (6.2) | 0.5 (2.6) |
| Row b. Maximum change in TTT (minutes) | | | | | | | | | | | | |
| Strike Φ [290:320] | 2 | 0 | 0 | 1 | 2 | 1 | 2 | 3 | 2 | 1 | 2 | 1 |
| Dip δ [25:60] | 6 | 2 | 2 | 4 | 2 | 2 | 5 | 6 | 4 | 3 | 3 | 5 |
| Rake λ [-120:-80] | 4 | 2 | 2 | 2 | 2 | 1 | 1 | 0 | 2 | 1 | 2 | 3 |

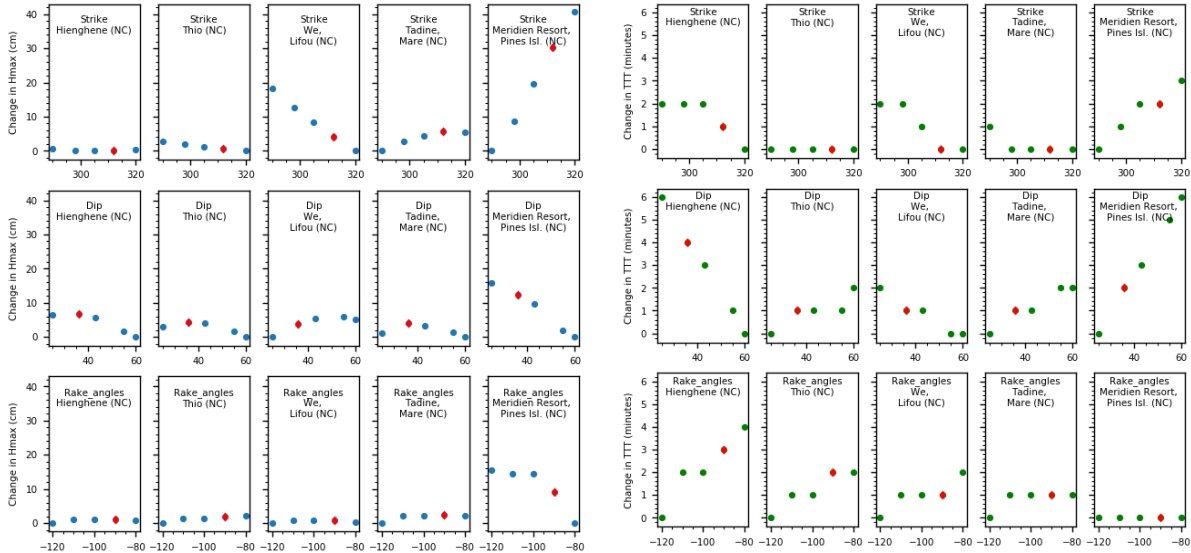

**Figure 9 : Examples of relationship between H<sub>max</sub> (left panel), or TTT (right panel) and the fault deformation angle (either strike or dip or rake). Results are extracted from the sensitivity tests at 4 places located in New Caledonia.**

## 6 Discussion

The comparison of the maximum energy path derived from the two scenarios (figure 6) and the sensibility tests shown on figure 9 highlights that UM exhibiting a 312° angle has a bigger impact on the Isle of Pines and Aneityum Island matching much better with the observations than NUM with an azimuth of 298°. In addition, the maximum wave height maps calculated over a high-resolution TIN grid (Figure 8) clearly indicate that the modelling results obtained with UM are in good agreement with the direct observations of the tsunami in both New Caledonia and Vanuatu. In fact, the coastal places where the modelling shows maximum amplitudes (> 0.4-0.5 m) are also the places where witnesses reported the tsunami (Isle of Pines, Aneityum, Yaté, Tanna, Erakor Island) and sometimes damages (Isle of Pines- Le Méridien resort, Aneityum, Mystery Island and southern coast to Umetch).

In addition, the tide gauge record comparisons show that globally the UM and therefore, the tsunami generation and propagation model, are together able to reproduce the tsunami records, in terms of arrival times especially in far-field location (Poindimié, Tanna and Port Vila tide gauges) (Figures 9e, g & h), polarity (Figures 9b, d, e, f, g & h), and amplitude (Figures 9b, e & h).

Except for Poindimié-Passe de la Fourmi where there is pressure sensor offshore the reef barrier, the observed delay between the simulations and the reality (the simulated signal being always the fastest) on all the New Caledonia coastal tide gauges managed by the SHOM (hydrographic service of the French navy) is mainly explained by the fact that there are some transmission issues from the gauge to the datacenter. Also, it has been demonstrated that the waves slow down during propagation due to reverse dispersions for the long periods for numerous reasons not considered in the presented modellings, leading to delays between the observed and simulated travel times up to 15 minutes for transoceanic tsunamis (Watada et al., 2014).

But small variations in fault orientation, like the dip for example, may also exert a control on the timing of the first leading wave in remote and shallow locations. As indicated in table 3, in row.b, places outside the lagoon

(Poindimié) or devoid of lagoon (Wé, Tadine) show little TTT sensitivity to dip variations, on contrary with Hienghène or Port Vila, indicating complicated interactions between changes in fault geometry and orientation parameters ($\Phi/\delta/\lambda$), seafloor details (like ridges and seamounts) and others geomorphological features (reef, lagoon, bay) on the tsunami wave propagation.

As a straightforward demonstration of the impact of both uncertainties in earthquake source parameters and influence of ridges on the wave propagation, two maps of $\frac{dH_{max}}{d\Phi}$ using slopes of the linear regression between $H_{max}$ and $\Phi$ are provided. In figure 10, left panel, the rugged seafloor of the Loyalty Ridge is simplified, with a flattening of shallow depths above 2500m (the flattened region is indicated on figure 5), while the original bathymetry is preserved in the right panel. From the map comparison, there is evidence that the Loyalty Ridge interacts with the tsunami waves at the first stage of propagation and that a part of tsunami energy is focused

onto the Loyalty Ridge by wave refraction. Similar mechanism of refocusing is at work along the eastern flank of the New Caledonia Ridge (Norfolk Ridge), trapping a portion of tsunami energy toward the Loyalty Basin. Finally, as pointed out earlier using the $H_{max}/\Phi$ relationship at Wé (Lifou), locations aligned with the rupture fault have a large sensitivity to bottom features, in particularly the northeastern shore of Maré.

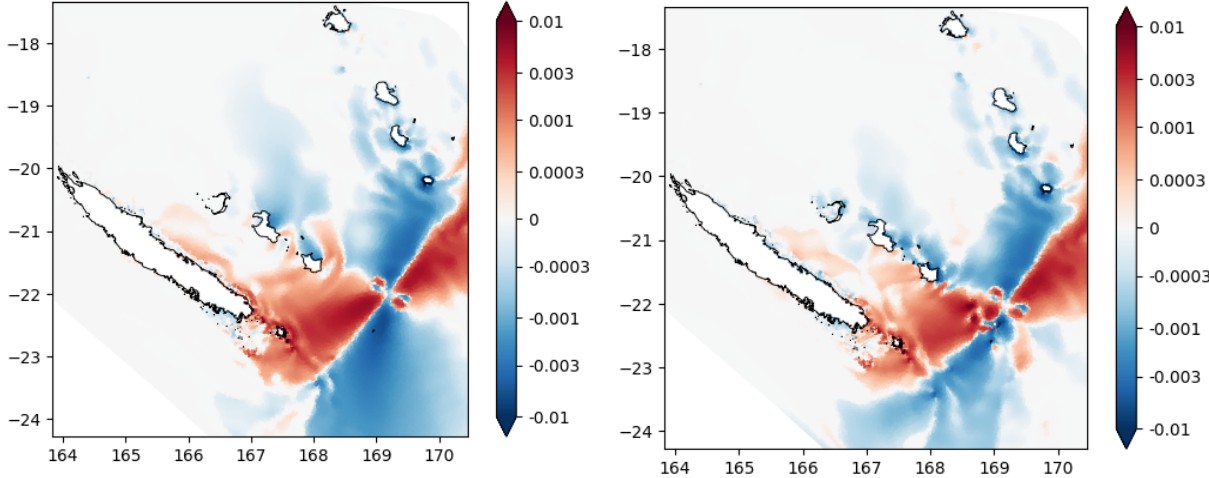

**Figure 10 : Spatial distribution of $\frac{dH_{max}}{d\Phi}$ across the model domain for the case with the simplified Loyalty Ridge (left**
**panel) ; the case with preserved bathymetry (right panel). Scale bar units are in cm/degree.**

Concerning the high frequency oscillations that the model is not able to reproduce, especially at Maré, Ouinné and Lenakel, it is presumably the result of resonant behavior of the tsunami waves interacting with semi-enclosed water bodies represented by Maré Harbor, Ouinné Harbor and Lenakel's Bay, and with the fringing reefs. A similar behavior has been described for other places in the literature (e.g. Horrillo et al., 2008;

Rabinovich, 2009; Aranguiz, 2015). The fact that the high-resolution coastal zones surrounding the location of the tide gauges have been built from sparse bathymetric data coming from low resolution nautical charts and aerial pictures interpretation could explain that the model is not able to reproduce the resonance as the shape of the water bodies and thus their natural oscillation modes are not exactly the same. According to previous studies, it is a safe bet that either a source refinement (complex source showing slip heterogeneity for example) or high-

resolution bathymetric data coming from multibeam or LIDAR surveys would be able to reproduce such phenomenon in these small and complicated places (e.g. Sahal et al., 2009; Vela et al., 2014).

Considering both maximum amplitude maps compared to the testimonials (locations and amplitudes) and the tide gauges simulation results comparison to the real recorded data, the simple fault plane rupture scenario chosen for this study provides quite good results compared to the more sophisticated one from USGS, based on

heterogeneous slip distribution. Observed and simulated TTT at Maré may suggest that the USGS fault geometry is inappropriate. This raises questions about their fault model inversion results for that event and a need to devote more effort in the settings of accurate earthquake fault model at the Loyalty Ridge-Vanuatu Arc junction.

It is interesting to notice that, nearly two years after the tsunami occurred, hidden observations are still

transmitted by witnesses. Tsunami modelling showing that the north and west coasts of the Isle of Pines would have also been impacted by the tsunami, several people were questioned during a field survey: a fisherman living at the Crab's Bay indicated that the sea receded from the bay and came back quickly in a rolling foam; the diving center and the Kodjeu Hotel located within the Ouaméo bay indicated that the diving club boat, supposed to be load at high tide, was laying on the sand instead at the exact arrival time of the tsunami (P.-E. Faivre, pers.

comm., 2020). Then the water came back and the sea rose above its natural maximum reaching the foot of the trees (according to a local fisherman, 2019), measured ~1 m above high tide.

**7 Conclusions**

Model results presented in this study for the December 5, 2018 Tadine tsunami indicate that using a simple fault plane rupture scenario is enough in such case of near field event to reproduce the tsunami correctly in terms of

maximum wave amplitude and polarity.

While there are some issues in simulated travel times, having serious implications for neighboring islands like Maré (TTT< 20 min), the more exposed places in New Caledonia (with Lifou and Ouvéa) to tsunami waves generated from the Vanuatu Subduction Zone, a probable origin may stem from inaccurate rupture parameters, like orientation angles, strike, dip and rake. The role of sharp changes in depth and tsunami wave refraction at

the crossing of the Loyalty Ridge raises the question of wave energy refocusing and trapping toward the Loyalty Basin, as demonstrated by flattening the local bathymetry. The question of possible wave amplification due to refocusing and reflection within the New Caledonia Archipelago will deserve future investigations using SCHISM, in order to increase our local knowledge on tsunami hazards for remote and sheltered locations.

In terms of study perspectives, it would be interesting to investigate how tides and lagoon hydrodynamics

interact with tsunami waves. The role played by the tide in tsunami impact has been demonstrated by several studies (e.g. Ford et al., 2014; Nakada et al., 2016). Such small amplitude event occurring at low tide could be dramatic as lots of people could be looking for shells and octopuses on the fringing reef.

Finally, considering the sea-level rise due to global warming in combination with storm surge or exceptionally high spring tides would also help to assess the future impact of small to moderate tsunami like the

December 5, 2018, over island communities with a question that arises: would the growth of coastal ecosystems such as corals and mangroves be able to adapt quickly enough to rising sea level to maintain their protective role against small events?

## Acknowledgements

The authors are very grateful to the Vanuatu Meteorology and Geohazards Department which provided the post tsunami survey report about Aneityum, and to all the people having shared their testimony of the December 5, 2018 tsunami collected within the months following the event, but also more specifically during the 2019 PALEOTSU field survey of paleotsunami deposits all around New Caledonia. They are also very grateful with Christopher Moore (NOAA) who provided support for the use of MOST and to Paul Wessel and Walter Smith who developed and maintain the free GMT mapping tools which was used to produce most of the maps of this study. Finally, they would like to thank Alberto Armigliato and an anonymous referee for their constructive comments to improve the quality of this paper. This study has been done within and funded by the TSUCAL project.

## Authors' contribution:

JR: study supervision; field investigations; DEM construction; numerical modelling; writing; figures preparation.

BP: study supervision; field investigations; writing; figures preparation.

MD: unstructured grid construction; data processing; figures preparation.

JL: numerical modelling; writing; figures preparation.

JA: funding acquisition; data processing; results discussion.

PL: seismic data processing.

BT: mapping; data processing.

CB: seismic network maintenance.

DV: seismic network maintenance.

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
