# Peer review of "The $M_w$ 7.5 Tadine (Maré, Loyalty Is.) earthquake and related tsunami of December 5, 2018: seismotectonic context and numerical modelling"

_Natural Hazards and Earth System Sciences, 2021_

## Referee Comment (RC2)

[referee-annotated manuscript omitted]

---

## Author Response (AR1)

Dear Editor,

We are delighted to send you a revised version of our manuscript firstly entitled "The Mw 7.5 Tadine (Maré, Loyalty Is.) earthquake and related tsunami of December 5, 2018: implications for hazard assessment in New Caledonia."

We have carefully considered all the comments provided by the two referees which helped to considerably improve the quality of our manuscript, making it more straightforward for the reader. Please find hereunder the check list of all the corrections/modifications related to the referees' comments and our answers in the discussion process. Note that some comments have already been answered in the discussion process and do not necessitate some corrections. Note also that, as the plan of the manuscript has changed, it was not possible to link the modifications to line numbers as you requested. Most of the modifications are visible in the version of the new manuscript showing the track changes.

Sincerely,

The authors.

**Referee #1**

*This paper presents an analysis of the Mw 7.5 earthquake and related tsunami occured on December 5th, 2018. The analysis included description of the source, field survey and tsunami numerical simulation. My major concern is that the paper does not have a clear focus and it is difficult to find one main idea throughout the paper. Is the focus the earthquke? the tsunami field survey?, the numerical simualtions? validation of numerical modes? or the analysis of tsunami hazard? I recommend to focus on the tsunami behaviour in order to explain why a small event can generate a large tsunami.*

- **This paper deals with the December 5, 2018 earthquake AND tsunami. It is important to discuss the complex tectonic background of the region first, as well as the history of earthquakes/tsunamis having affected New Caledonia. Also, working on tsunami modelling doesn't prevent to relate to reality and deal with field observations. This event, even if small in comparison to other tsunamis in other regions of the World, is of main concern for New Caledonia, Vanuatu and nearby countries and all the aspects must be presented to understand the issues.**
  - ➔ The title and the plan of the new version of the manuscript have been changed, making the demonstration clear for the reader: the new title is no more referring to hazard assessment in New Caledonia; the tectonic context of the region has been moved from the introduction in a specific part called "Seismotectonic context"

*Then, the field survey, tide gauge data and numerical simulations can be used to explain that phenomenon. I do not believe that with one event you can make an analysis of the implication for tsunami hazard. It could be mentioned in the discussion, but as the paper is right now, it is not possible to include it in the title, unless the focus of the paper is the tsunami hazard. In*

*that case, more information and state of the art on tsunami hazard in New Caledonia should be mentioned in the Introduction as well as your current contribution.*

- **The paper does not focus specifically on tsunami hazard assessment but presents how one event, which shows quite important tsunami amplitudes (comparing to the history of tsunamis in New Caledonia) and is the only tsunami recorded on all local tide gauges, can be useful to validate the way the tsunami hazard assessment is/will be done for the archipelago. That's why we used "implications for tsunami hazard assessment" in the title, to provide information to the reader how one event like this can help in a place where so few events are available, because of very recent history (< 150 years).**

    - ➔ The title has been changed to "The Mw 7.5 Tadine (Maré, Loyalty Is.) earthquake and related tsunami of December 5, 2018: seismotectonic context and numerical modelling" in order to avoid confusion with detailed hazard assessment studies.

*Another concern is related to tsunami numerical simulations. Only uniform slip distribution is used, but it has been demosntrated that uniform slip may underestimate the hazard. See for example Melgar et al 2019; Carvajal and Gubler 2017; González et al. 2020; Geist and Dmowska 1999. This could be an explanation why simulated tsunami waveforms at tidegauges are not in agreement with tide gauge records Would it be possible to use a nonuniform slip distribution model? It would be desirable to propose some finite fault models of the earthquake by means of seismic records, thus seismic parameters such as strike, dip and rake are properly defined.*

- **We have tested the heterogeneous slip distribution provided by the USGS (https://earthquake.usgs.gov/earthquakes/eventpage/us1000i2gt/finite-fault) and because there are no substantial differences with the uniform slip distribution on the tide gauge records we had decided not to show the results in the paper. In this particular case, the uniform slip distribution shows higher amplitudes than the non-uniform one on tide gauges. Also, the solution proposed by the USGS does not fit at all with the empirical relationships between earthquake parameters and seismic moment proposed in the literature for such rupture (Strasser et al., 2010; Blaser et al., 2010, etc.).**

    - ➔ New simulations have been done using the USGS non-uniform slip model and the results have been integrated in the paper. This solution is definitely not enough to reproduce the tsunami records. We also discuss the fact that the finite source from USGS is not correct in terms of geology (too long rupture for such magnitude, azimuth not in agreement with geological structures).

*The paper does not have an introduction, and the main objective of the paper is not clearly described. In addition, the section 1 "General Setting" is too long to be an introduction. this section should be compressed and be part of a "Study area" section. It is very important to define the focus of your paper, then, define which information regarding the tectonic setting is relevant.*

- **We agree that the introduction is long, but it is important to set up correctly the complex tectonic background of the region as well as dealing with the historical**

**events especially for people not familiar with this region. We propose to change the plan of the manuscript a bit, moving that information about tectonic background in a dedicated part, reducing the size of the introduction and making the focus of the paper clearer.**

➔ The introduction and the plan itself have been changed, with a new part dealing specifically with the tectonic settings.

*The title mentioned the "implications for tsunami hazard", but none of the sections mentioned anything realted to current tsunami hazard analysis in the study area. If the focus of your paper is the implications for tsuami hazard, you should present the state of the art of tsunami hazard in New Caledonia, such that the scientific gap and your contribution are clearly described in the introduction.*

- **Not necessarily: the implications could be for a future study of tsunami hazard assessment, for example, how we plan to use this event as a basement for updating the actual scenario database.**

➔ This part of the title has been removed to avoid confusion.

*Another minor comments are the following*

*- Figure 1. Please add a general map here, such as the left hand side map in Figure 3.*

- **We will add a general map here as requested on an updated version of the figure.**

➔ Map locating New Caledonia in the Pacific Ocean has been added as an inset on the figure.

*- line 46, it says "pressure gages", but in other places, such as line 425 it says "tide gauges". Use gage or gauge, but not both.*

- **This is a typo mistake, it should be "gauge"**

➔ there was only one mistake and we have corrected it.

*- section 1.1 and 1.2 report significant amount of information regarding past earquakes, but the point is not clear. Would it be possible to combine Figure 1 and 2 and summarize the important facts only?*

- **Historical seismic activity of the region is shown on Figure 1 with the identification of events who generated a tsunami recorded in New Caledonia. Figure 2 is focused on the different seismic crises which occurred in the specific area of the 5 December earthquake and illustrated the special tectonics of the area. Figures 1 and 2 are complementary.**

➔ Figure 1 and 2 haven't been combined because we find it more interesting to detail each past seismic crisis but the text has been improved and sentences about past earthquakes reworked, especially according to referee #2 comments.

*- line 161 indicate a finite fault model from USGS. Why didnt you use this model? You only mentioned this model, but do not explain why it is discarded.*

- **This model has been used and compared to the other simulations but does not provide better results. The objective was to show that a simple fault plane with uniform slip can produce results in the same order than the observations/records in an operational aspect. Anyway, we understand that after more than 2 years since the earthquake occurred, people would like to see results obtained with more detailed sources and thus, it will be added in the new version of the manuscript.**

    ➔ The finite fault model from USGS is now incorporated in the manuscript as one of the two main scenarios.

*- Figure 3. It is difficult to read this figure. I recommend to improve it, for example, add also field survey data to this figure, and add the magnitude of the measurement (the number) and not only a color. Instead of a color, add insets with the tidegauge records, thus we could see maximum amplitudes and the tsunami behavior as a function of time.*

- **We agree that the figure 3 is a bit difficult to read and should be improved; it is a good idea to add the field survey/observation values to the tide gauge maximum amplitudes. Concerning the tide gauge records they are already shown on figure 8.**

    ➔ The figure has been improved according to the referee's suggestions. We haven't added the tide gauge records as insets because they are already shown on figure 7 (comparison with simulated time series).

*- Section 2.2.2. All measurements and maximum amplitudes reported in this section should be listed in a table with longitude and latitude coordinates. In addition, measurementes should be indicated in Figure 3, as mentioned in a comment above.*

- **We agree that a table would be more efficient for the readers; it will be added in the new version of the manuscript.**

    ➔ Finally, we have chosen not to add the requested table as it has now been added to the figure 3.

*- Fiure 5, please draw the coastline to see better the islands.*

- **The coastline will be added in the new version of the figure.**

    ➔ Coastline added to the figure.

*- line 292. Please explain how L=80km and W=30km were defined. The model from USGS used L=160km. How did you come up to 80 km? did you use a scale relationship? if so please explain which ones.*

- **The values have been obtained using the empirical relationships from Strasser et al. (2010) and Blaser et al. (2010). To fit the results of these 2 studies, and in**

**agreement with the geological/tectonic context of the region, a Mw = 7.5 normal faulting earthquake could be associated to a rupture showing length L~80 km (maximum value obtained with all the relationships) and width W~30 km (minimum value obtained with all the relationships). We agree that the paragraph dealing with this must be included in an update of the manuscript.**

➔ Explanation added to the manuscript.

*- lines 285 to 295. Several seismic parameters are presented here, however, in line 353 two strike anlges are mentioned, but they were not mentioned earlier. I suggest to add a table in section 3.1.2 with all seismic parameters used in the numerical simulations, including the two strike angles and depth.*

- **We agree with this remark and must add a table gathering all the parameters in a further version of the manuscript.**

  ➔ Table has been added as requested.

*- lines 336 to 340. You described a sensitivity analysis regarding the kinematic tsunami initial condition. I understand that you used a linear variation during 50s with time step of 1s, but previously you mentioned that the USGS model has 3 patches and it website shows a rate of moment release with 4 peaks during the 50 s. Therefore, it is inconsistent.*

- **It is right. Since this remark, we managed to simulate the complex and dynamic seabed deformation as suggested from the USGS dynamic rupture model and did not see any substantial change compared with the USGS static model.**

  ➔ This information has been added to the manuscript.

*Please explain why you made a sensitivity analysis. Why not simply using a static sea bottom deformation by means Okada formulation?*

- **There is no sensitivity analysis about the question of how to model the deformation since we use the default behaviour as implemented in schism: a rupture from the seabed propagating through the water column. Seabed motions are driven by the Okada's solution.**

  ➔ the paragraph dealing with the seabed initial deformation has been reworked to make it clearer.

*- line 353. Please explain why you used these two angles in section 3.1.2.*

- **Explained in the text (see line 158 to 160): the strike from USGS and GCMT was not the same and we have chosen the GCMT solution because it is more in accordance with the geological knowledge of the region (bathymetry, identified structures, etc.).**

  ➔ The former USGS scenario has been replaced with their finite model and we discuss the problem of the azimuth of this source in the manuscript.

*In addition, in line 159 you mentioned several combinations of parameters according to different observatories, but later you select only one dip and rake ablges, and vary the strike angle. Please explain better the assumtions in section 3.1.2. from lines 291 I understand that you will use the GCMT parameters (312°, 36° and 90°) but then you also used 298° for strike angle.*

- **We only show one of the multiple sensitivity tests we have run during the process. This one produced the most important variation of main energy path orientation and the maximum amplitude on the tide gauges.**

    ➔ The sensitivity tests have been added and are now discussed in the manuscript.

*- line 342. It says that simulations have a slapsed time of 3 h, but it may be short considering the distance from the source to the points of interest and resonance effects, as mentioned in line 250. Did you check any resonance effect?*

- **Yes we did. It does not provide any interesting results.**

    ➔ This is now discussed in the manuscript.

*- Figure 6. I suggesto to draw the trech in all figures in order to better sea differences between tsunami source models.*

- **This will be added in a new version of the manuscript.**

    ➔ The trench has been added.

*- lines 372-420. results here are only descriptives, and no analysis nor discussion on tsunami behavior is presented. Please, according to your results and analysis, explain why you have amplification (see lines 378, 379, 383, etc).*

- **The discussion of the results is provided in the Discussion section (part 4) between line 427 and line 463.**

    ➔ Discussion section is now part 6 and has been improved.

*- lines 396 to 399. It says that the modelling is not able to reproduce the tide gauge record in terms of arrival time and amplitude, but no analysis nor discussion are presented in order to find possible causes. Please explain possible causes such as grid resolution, bathyemtry errors, fault location (fault model is closer to the tide gauge than the real rupture), uniform v/s heterogeneous slip, etc?*

- **As said above discussion is presented in Discussion section (part 4).**
- **After revisiting the nonuniform and dynamic USGS model, we also noticed a possible degradation in arrival times compared with our custom fault model, which is more consistent with the local tectonic context. Impacts of fault location will be discussed in the revised version of the manuscript.**

    ➔ This is detailed in the manuscript.

*- Figure 8. only two hours of numerical simulations are shown. If you simulated 3h, please show the 3 h. It seems that the third wave in figure f is larger than the second, but it is not shown. In addition, adjust the vertical scale in figures g an h, thus the whole amplitude is shown in the plot.*

- **We agree with this comment and we will show the results until 3 hours in a new version of the manuscript. Vertical scale will also be fixed.**

  ➔ The figure shows now 3 hours of tsunami propagation and the vertical scale has been fixed.

*- The paper needs a Discussion section, thus all results and implication in tsunami hazards are discussed. In addition, you can discuss whether the phenomena observed here (amplification, defocusing, resonance, etc) have been observed in other places, thus you can explain the tsunami bahaviour in the current event, and , hopefully, explain what could happen in future events.*

- **Discussion section available (part 4) between line 427 and line 463.**

*- line 467-468. it says "to reproduce the tsunami correctly..." I am not sure whether your simulations allow to conclude this. I rather say this is in fair agreement, since most of your tsunami waveforms are not well reproduced. If you only compared maximum amplitudes, this analysis whould be included in a discussion section. You can also compare numericaly the simulated and measured maximum amplitudes.*

- **We just said "that using a simple fault plane rupture scenario is enough in such case of near field event to reproduce the tsunami correctly with a hazard management point of view ». In the operational context of tsunami hazard management in New Caledonia, and considering the complexity of the coastline bordered by both a fringing reef and a barrier reef, one of the objectives was to test the simple process used to prepare a scenarios database.**

  ➔ Conclusion about the results has been improved.

*- lines 469-470. Is the paper focused on the validation of MOST and SCHISM models? I am not sure if you can conclude this, due to the fact that arrival time and maximum amplitudes given by your simualtions do not show good agreement. Possible errors (epistemic and aleatory) are many, and you can also discuss about this in a Discussion section.*

  ➔ The reference to MOST was a mistake and has been removed. The discussion of the results has been improved, and we do not "validate" SCHISM anymore with only the modelling of one solely event.

*- lines 473-477. These statements are more suitable for a discussion section.*

- **Answer for the 2 last questions/comments: Discussion section available (part 4) between line 427 and line 463.**

*- Finally, the conclusions are very weak, since there is no Introduction section with clear objective and there is no Discussion section either.*

- **The introduction will be shortened and clarified with a specific part for the objective and a part for the tectonic context; the available discussion section (line 427 and line 463) will be improved in a further version of the manuscript.**

  ➔ Introduction has been shortened and discussion improved.

**Referee #2**

*The first point regards the representation of the earthquake fault. One of the main conclusions drawn by the authors is that "using a simple fault plane rupture scenario is enough in such case of near field event to reproduce the tsunami correctly with a hazard management point of view". What is the tolerance that authors adopt to consider correct the event's reconstruction they present? To what extent a systematic time-advance in the tsunami arrival time simulation, a significant underestimation of the maximum amplitude and an overestimation of the wave period at some coastal sites can be considered acceptable? Have these aspects been investigated more in detail by taking into account at least one possible heterogeneous slip distribution on the fault? I see two possibilities: the simulation of the tsunami obtained taking into account the slip heterogeneity can either improve the results regarding at least one of the problematic aspects listed above: in this case, the authors should point this out and discuss the possibility to introduce some form of slip heterogeneity in the hazard assessment procedure;*

- **We have tested the heterogeneous slip distribution provided by the USGS (https://earthquake.usgs.gov/earthquakes/eventpage/us1000i2gt/finite-fault) and because there are no substantial differences with the uniform slip distribution on the tide gauge records we had decided not to show them in the paper. We agree that it could be interesting to show them, what would help to improve the discussion.**

  ➔ We have finally integrated the modelling of the finite fault model as indicated hereabove because it shows the weakness of this source, not being able to reproduce the records/observations at all, and being far from the geology of the region (length of the fault not correlated with such magnitude, azimuth which is not related to existing features).

*or*

- *introduce no significant improvement in any aspect of the tsunami simulations: in this case, the authors can safely confirm their conclusion, but this must be supported by concrete results.*

*Still regarding the parameterization of the earthquake fault, the role of the strike is investigated by taking into account two of the early strike solutions provided by seismic networks. The effect on the tsunami simulations is illustrated only by means of maximum energy distribution maps. But what about the tide gauge records?*

- **The tide gauge records of the 2 different strike presented in this study have been compared and show differences between the 2 cases as expected; nevertheless, we have decided to show only the one fitting the best with the observations and tide gauge records; we can also add it to the manuscript for clarity.**

  ➔ Synthetic records for the two scenarios are compared to the real records and help to eject the USGS scenario and then focus on our simple fault model.

*Moreover, how can the information deduced from the comparison be translated into suggestions for the hazard assessment procedure?*

- **The actual database built to assess tsunami hazard in New Caledonia and help decision-makers to evacuate the coastal areas or not, is composed of more than 3000 scenarios located all around the Pacific Ocean. In case of an earthquake, the pre-computed maximum wave amplitude maps from the closer scenario to the epicentre are selected. This scenario has specific parameters which are following the global shape of the subduction zone (strike, dip, rake, coupling width). The comparison of the different results obtained with different strike (but also dip, rake, etc., not shown in the manuscript) highlight the necessity to complete the database with additional scenarios, based on very detailed analysis of the seismicity and the geological features in this complex region.**

  ➔ We have decided to remove any mention to hazard assessment in New Caledonia as the database as still not be published.

*Concerning the tsunami modelling part, the authors mention that a 7-km resolution regular grid is used mainly to model the generation process. How is this grid matched with the unstructured grid?*

- **In details, the static seabed deformation is treated like an anomaly in the MOST deformation module based on Okada's solutions and this anomaly is transferred from a 7-km regular grid to the unstructured SCHISM grid. As a side note, in the region of the fault, the same bathymetric data (Smith and Sandwell) is shared in both the 7-km grid and unstructured grid.**

  ➔ The methodology has been revised and MOST is no more considered in the process: all the stages are done using SCHISM from seabed deformation to tsunami propagation over only one unstructured grid (TIN DEM in the text).

*For the 7-km grid, why was the Smith and Sandwell (1997) database used instead of more recent databases (for instance GEBCO_2020)?*

- **The high resolution DEM used for the modellings has been prepared in the aftermath of the event in early 2019; the last version of GEBCO hadn't been already released. Also, comparison between the GEBCO 2014 and Smith and Sandwell (1997) dataset for what concerns New Caledonia territorial waters highlighted some strong differences between the two and artefacts (like unreal seamounts) mainly present on GEBCO grid. Those reasons explain the use of Smith and Sandwell data in this area (only for filling in the deep-water parts not covered by high-resolution multibeam data).**

➔ We don't find it relevant to deal with this in the new version of the manuscript.

*The SCHISM code is a feature-rich tool that appears to be used in the paper as a nonlinear shallow water code. Is this the way it is foreseen to be used also in the future hazard assessment strategy?*

- **SCHISM has been used to produce more than 3000 scenarios for tsunami warning for the Civil Defence Office of New Caledonia and tsunami hazard assessment purpose. It has run through the different benchmarks commonly used for tsunami model validation (https://nctr.pmel.noaa.gov/benchmark/index.html).**

*I think the authors should elaborate further their conclusion that the time shift observed in the tide-gauge records between simulations and observations is imputable to "transmission issues from the gauge to the datacenter". What kind of issues are we talking about? Are these issues present only for the New Caledonia stations? How to justify the advance in simulated arrival times for the other tide gauge records? Wouldn't it be useful to play a bit with the fault geometry and position to see how the comparison changes?*

- **Multiple tests have been run to try to fix this "shifting" problem: different fault geometries, multiple segments, slip distribution, local improvements of the bathymetric data have been tested but none ended up on better results: the problem comes from the fact that there is an identified shift due to transmission issue as indicated lines 440 to 443.**

  ➔ Discussion has been improved following this comment.

*Concerning the introductory "General setting" chapter, I think it is much longer than needed. Only a little part of the wealth of information provided in that chapter is useful in the following discussion. I strongly recommend to shorten this part keeping only the information that is useful for the subsequent discussion.*

- **We agree that the introduction is long, but it is important to set up correctly the tectonic background of the region as well as dealing with the historical events. To make it clearer, we could change the plan of the paper a bit, moving that information in a dedicated part.**

  ➔ As indicated in the referee #1's part, the plan of the manuscript has been revised and the introduction part has been considerably reworked, extracting the tectonic context toward a specific part.

*The style is sometimes cumbersome, with several repetitions in some places. Formatting (especially regarding figure captions and references in the text) should have been checked before submitting the paper.*

- **We don't understand the issue: indeed, everything has been carefully checked before submitting, and neither the PDF version we collected after the submission process nor the one still available online (https://nhess.copernicus.org/preprints/nhess-2021-58/nhess-2021-58.pdf) show the same formatting issues as the one you sent as supplementary comments.**

*I am attaching an annotated version of the paper, containing several corrections and suggestions for improvement.*

- **Thank you for those additional comments which will be considered in a further version of the manuscript.**

  ➔ All the corrections suggested by the referee have been included in the revised version of the manuscript.